# Computational memory capacity predicts aging and cognitive decline

Mite Mijalkov [1] ✉, Ludvig Storm[2], Blanca Zufiria-Gerbolés [1], Dániel Veréb [1], Zhilei Xu [1], Anna Canal-Garcia [1], Jiawei Sun[1], Yu-Wei Chang[2], Hang Zhao[2], Emiliano Gómez-Ruiz[2], Massimiliano Passaretti[1], Sara Garcia-Ptacek[3,4], Miia Kivipelto[3,5], Per Svenningsson [1], Henrik Zetterberg [6,7,8,9,10,11], Heidi Jacobs [12,13], Kathy Lüdge [14], Daniel Brunner [15], Bernhard Mehlig [2], Giovanni Volpe [2] ✉ & Joana B. Pereira [1] ✉

Memory is a crucial cognitive function that deteriorates with age. However, this ability is normally assessed using cognitive tests instead of the architecture of brain networks. Here, we use reservoir computing, a recurrent neural network computing paradigm, to assess the linear memory capacities of neural-network reservoirs extracted from brain anatomical connectivity data in a lifespan cohort of 636 individuals. The computational memory capacity emerges as a robust marker of aging, being associated with resting-state functional activity, white matter integrity, locus coeruleus signal intensity, and cognitive performance. We replicate our findings in an independent cohort of 154 young and 72 old individuals. By linking the computational memory capacity of the brain network with cognition, brain function and integrity, our findings open new pathways to employ reservoir computing to investigate aging and age-related disorders.

The human brain is a complex system of anatomically interconnected regions. The resulting network, the connectome[1], is responsible for the transmission of signals across different brain regions, allowing them to interact and exchange information[1]. These dynamic interactions generate complex patterns of functional activity that are closely associated with our behavior and cognitive functions[2,3]. In young healthy individuals, such interactions feature a non-random efficient organization, which optimally balances the network's ability to process information through a small number of designated paths between distant regions on the one hand, and many local connections between neighboring regions on the other hand, achieving a delicate balance between global and local information processing[4]. However, this optimal organization becomes impaired as aging progresses[5,6], being associated with a decreased capacity to efficiently transmit information, as well as with a decline in the number and integrity of anatomical connections[7–9]. Altogether, such changes hinder the communication between brain regions, ultimately contributing to the cognitive decline associated with aging[6,9].

[1]Department of Clinical Neuroscience, Division of Neuro, Karolinska Institutet, Stockholm, Sweden. [2]Department of Physics, Goteborg University, Goteborg, Sweden. [3]Department of Neurobiology, Care Sciences and Society, Division of Clinical Geriatrics, Karolinska Institutet, Stockholm, Sweden. [4]Theme Inflammation and Aging. Aging Brain Theme. Karolinska University Hospital, Solna, Sweden. [5]University of Eastern Finland, Kuopio, Finland. [6]Department of Psychiatry and Neurochemistry, Institute of Neuroscience and Physiology, the Sahlgrenska Academy at the University of Gothenburg, Mölndal, Sweden. [7]Clinical Neurochemistry Laboratory, Sahlgrenska University Hospital, Mölndal, Sweden. [8]Department of Neurodegenerative Disease, UCL Institute of Neurology, Queen Square, London, UK. [9]UK Dementia Research Institute at UCL, London, UK. [10]Hong Kong Center for Neurodegenerative Diseases, Clear Water Bay, Hong Kong, China. [11]Wisconsin Alzheimer's Disease Research Center, University of Wisconsin School of Medicine and Public Health, University of Wisconsin-Madison, Madison, WI, USA. [12]Maastricht University, Maastricht, Netherlands. [13]Massachusetts General Hospital, Boston, MA, USA. [14]Institute of Physics, Technische Universität Ilmenau, Weimarer Straße 25, Ilmenau, Germany. [15]Institut FEMTO-ST, Université Franche-Comté, CNRS, Besançon, France. ✉e-mail: mite.mijalkov@ki.se; giovanni.volpe@physics.gu.se; joana.pereira@ki.se

Until now, the precise impact of these changes on the computational capacity of the human brain to process, learn, and to encode temporally varying external stimuli has remained elusive. Here we use reservoir computing[10]—a machine-learning framework that can reproduce the functional dynamics of biological networks[11–14]—to characterize how the cognitive function of the human brain deteriorates with aging.

A reservoir computer is a recurrent neural-network model. It consists of a reservoir of artificial neurons, which are nonlinear computational nodes[10]. The reservoir receives inputs in the form of time series, and the network output is trained to encode different representations of these signals. The linear memory capacity[15,16] of this model measures how well the network manages to encode the input signal in its reservoir. This *computational* memory capacity is determined by training the network output to reproduce delayed input time series and comparing the delayed input with the reservoir output, either across all reservoir nodes (*global* memory capacity) or across a select subset (*regional* memory capacity). A high memory capacity indicates that the network has high capacity to remember and process the temporal information contained in the input signal.

Here, we evaluate the computational memory capacity of anatomical brain connectomes in a lifespan cohort of 636 individuals between 18 and 88 years old. We model the reservoir nodes to represent different brain regions and constrain their inter-regional connectivity by the individual anatomical brain connectomes using diffusion-weighted imaging. Then, framing the connectomes as computational reservoirs, we examine their ability to replicate random time-dependent input signals. We determine how the computational memory capacity of such brain-inspired reservoirs depends on the connection densities of the underlying connectomes and find that there is an optimal density to predict an individual's age. We replicate our findings in an independent cohort of young and old individuals. To assess the clinical implications of our findings, we test whether the computational memory capacity is associated with cognitive performance and different brain imaging measures derived from gray-matter atrophy, anatomical and functional connectivity, as well as the integrity of locus coeruleus, a small subcortical nucleus that is responsible for most of the noradrenergic neurotransmission in the brain. Our underlying hypothesis is that the computational memory capacity of the anatomical network is associated with individual cognitive performance and neuroimaging measures of brain structure and function. Our analysis shows that the computational memory capacity is a sensitive imaging biomarker for predicting age-related changes, with important implications for understanding, and potentially diagnosing and treating neurological disorders.

## Results

### Evaluation of the brain memory capacity

To test this hypothesis, we measured the memory capacity of 636 individuals from the Cambridge Centre for Ageing and Neuroscience (Cam-CAN) cohort[17] (https://www.cam-can.org) who underwent diffusion-weighted imaging (DWI) and neuropsychological testing. For each individual, the whole-brain anatomical connectivity networks were obtained by applying deterministic streamline tractography to multi-shell diffusion-weighted data and extracting the mean fractional anisotropy (FA) values between each pair of brain regions from the automated anatomical labeling atlas[18] in order to define the edge's values. Since the resulting networks can have a different number of connections (network density) across individuals[19], and given there is currently no consensus on which density is best[20], we performed our analysis across a range of network densities to ensure a meaningful comparison across individuals at different ages in which all connectomes had the same number of connections[19,20].

To measure the computational memory capacity of the individual connectomes, we used reservoir computing to simulate a representation of the propagation of random time-dependent signals through the anatomical connections of the brain of each subject (Fig. 1A). After letting this signal propagate through the reservoir (representing the anatomical brain networks) and activate its nodes (representing the brain regions), we recorded the activity series of the brain regions (the output signal in Fig. 1A) and fed them to a linear unit that was subsequently trained to reproduce a delayed representation of the input signal (Fig. 1B). We evaluated the global memory capacity of the whole network by using the activation time series of all reservoir nodes, whereas we defined the regional memory capacities as the ability to reproduce the delayed input signal using only the activation time series of single brain regions. The validity of this method to measure regional memory capacity was confirmed by an additional analysis we performed (lesional memory capacity; Supplementary Methods 1 and Supplementary Fig. 1), which quantifies the memory capacity of a region as the change in the network's global memory capacity after the removal of that region and all its connections from the network.

For each time delay $\tau$, we plotted the delayed representation of the input signal at that delay versus the corresponding prediction signal made by the reservoir. Such representations are illustrated in Fig. 1C for delays of $\tau = 1$ (red) to $\tau = 35$ (gray), where the delays denote the number of steps used to construct the representation of the input signal to be memorized. The degree to which the reservoir prediction approximates the delayed representation signal can be calculated using the Pearson's correlation coefficient between the two signals; the reservoir's forgetting curve (Fig. 1D) summarizes these correlation coefficients as a function of the time delay.

The memory capacity is calculated by summing the correlation coefficients in the forgetting curve across different delays, which reflects the total ability of the nodes to encode the temporal properties of the external stimuli[15]. The reservoir's forgetting curve (Fig. 1D) demonstrates that the network almost perfectly memorizes the input signals at small delays (between 1 and 5). We focused our analyses on the second part of the curve and evaluated the computational memory capacity at delays ranging between 6 and 35, since the network's ability to perfectly memorize the signal at small delays did not change with aging (Supplementary Methods 2, Supplementary Figs. 2 and 3), while the network was unable to memorize the input signal at larger delays.

### Memory capacity during aging

**The connectome of older individuals shows lower computational memory capacity.** We assessed the relationship between memory capacity and age by dividing the cohort using the mean age of the sample into two groups, young (18–53 years old) and old (54–88 years old) individuals, and comparing their memory capacity across different network densities using nonparametric permutation tests. Figure 1E shows the differences between the two groups as a function of density and the corresponding confidence intervals. It shows a significant decrease in the network's memory capacity in the older compared to younger individuals across the entire density range ($p_{min} < 10^{-4}$). Interestingly, these differences were smaller at low densities when only the strongest 10% connections were retained in the network, but progressively increased with higher network density, when weaker connections were additionally included in the network (Fig. 1E). The differences had a large effect size (Cohen's $d > 0.8$) and were even more pronounced when comparing more extreme age groups, i.e., with increasing age gaps between the two groups (Supplementary Methods 3 and Supplementary Fig. 4). Furthermore, they remained unchanged after changing the threshold of 53 years by several different thresholds (39–70 years) to place individuals in the young and old groups (Supplementary Methods 4 and Supplementary Fig. 5).

To confirm the findings showing that the memory capacity decreases in old individuals, we divided the complete density range

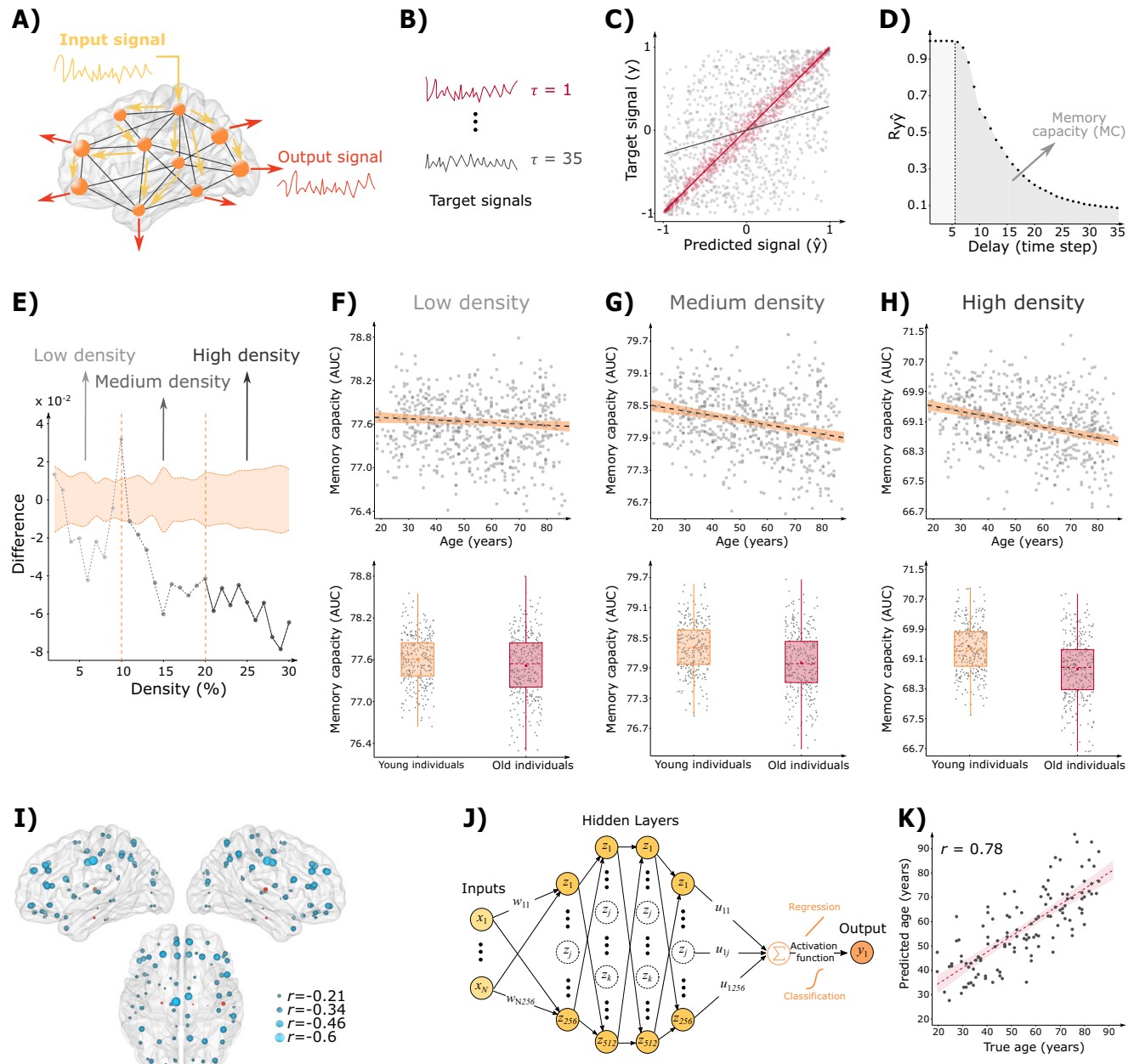

**Fig. 1 | Memory capacity in aging. A** Brain reservoir computing architecture with uniform random signals applied to all nodes. We recorded the reservoir's output as an activity series for the calculation of global and regional memory capacity. **B** Using this output, we trained the reservoir to reconstruct delayed representations of the input signals ($\tau = 1 - 35$). **C** Similarity between the reconstructed and actual input signals was measured using Pearson's correlation coefficient for each $\tau$. **D** Pearson's correlation coefficient as a function of $\tau$; the memory capacity was measured between $\tau = 6 - 35$ (dark-gray area). **E** Differences in global memory capacity between older (54–88 years, $n = 333$) and younger (18–53 years, $n = 303$) individuals. Gray dots show differences as a function of network density; 95% CI are shown in orange. **F–H** Correlation between global memory capacity and age at low, medium, and high densities, respectively. In all cases, y-axis shows AUC values, gray dots indicate individual subjects, dashed lines represent best model fit, and orange areas show 95% CI. Boxplots display AUC of the global memory capacity values for young ($n = 303$, orange) and old ($n = 333$, red) subjects. They show the median (center line) and the mean (colored center circles) of each group, the box boundaries extend to the 25th and 75th percentiles of the sample with whiskers indicating non-outlier data. **I** Regional memory capacity as a function of age, with negative correlations in blue and positive in red. Only regions passing correction for multiple comparison ($FDR < 0.05$) are shown. Brighter colors and larger spheres indicate higher absolute correlation. **J** Multilayer perceptron trained to predict age using global and regional memory capacity. **K** Predicted vs. true age for the best prediction (correlation coefficient 0.78); dashed line represents best model fit, shaded red area indicates 95% CI for predictions, and gray dots denote individual subjects on the test set. The brain surfaces are based on the MNI152 template[93–95]. Source data for Fig. 1 is provided as a Source Data file.

into low (2–10%, Fig. 1F), medium (11–20%, Fig. 1G), and high (21–30%, Fig. 1H) densities and calculated the area under the curve (AUC) by integrating the memory capacity values across all density values in each category. These AUC values were then used as dependent variables in separate linear models, with age as the independent variable and sex as a covariate. The analyses showed that age was the best

predictor of memory capacity at high densities (10.9% explained variance, $p_{age} < 10^{-15}$), surpassing the predictive power of the models using low (3.98% explained variance, $p_{age} = 0.04$) and medium densities (6.87% explained variance, $p_{age} < 10^{-12}$). The high-density connectomes consist of the strong connections present at low and medium densities, added to a set of weaker connections that are present at high

densities, but not at low and medium densities. Therefore, these findings show that the age-related reductions in memory capacity are most evident when an increasing number of weak connections are present in the network. This suggests that the more peripheral network connections of the network are most vulnerable to age-related effects, in line with previous studies[5,21]. Motivated by these interesting findings, we focused the rest of our analyses only on the high-density memory capacity. Moreover, we conducted additional analyses to assess whether topological measures or FA values explained the relationship between memory capacity and age. Our findings showed that, while these measures partially mediated the relationship, memory capacity provided independent information about connectome changes throughout aging (Supplementary Methods 5,6 and 7, Supplementary Figs. 6,7 and 8).

**Computational memory capacity of frontal and parietal regions declines with aging.** We investigated the impact of aging on the memory capacity of individual brain regions by assessing the correlation between regional memory capacity and age, using sex as a covariate. After controlling for multiple comparisons, our findings showed that most regions exhibited a decreasing memory capacity with age, with the strongest decline being observed in lateral frontal regions, followed by the middle and anterior cingulate, the precuneus, and the inferior parietal gyri (Fig. 1I). These regions overlap with some of the most connected brain areas of the default-mode network[22], and therefore receive a high number of weak connections with increasing network density. Interestingly, we also observed that a few regions showed a positive correlation with age, such as the Heschl's and parahippocampal gyri, potentially indicating a compensatory phenomenon that occurs during clinically normal aging as evidenced by previous reports showing such effects in these regions as a mechanism to cope with changes in other brain areas[23–26].

**Computational memory capacity is an accurate predictor of individual age.** To establish the ability of global and regional memory capacity to predict age, we built a multilayer perceptron deep learning model[27] (Fig. 1J) with four hidden layers (with 256, 512, 512, and 256 artificial neurons). More details about model selection are shown in Supplementary Methods 8 and Supplementary Table 1. An unbiased age prediction was ensured by subdividing the dataset into train ($n = 508$), validation ($n = 64$) and test ($n = 64$) sets with similar age distributions. The model performance was evaluated by calculating the Pearson's correlation coefficient between the true individual age and the predicted age, where a higher correlation indicates a better model performance. Implementing a 10-fold cross-validation procedure, we found that this model showed a high correlation between the true and predicted individual age of 0.72 ± 0.04, where the best prediction is shown in Fig. 1K. This demonstrates that memory capacity can be used to make highly accurate predictions of an individual's age, being therefore a robust imaging biomarker that tracks age-related changes.

**Associations between memory capacity, neuroimaging measures of brain structure and function, and cognition**
**Global and regional memory capacities are associated with measures of structural integrity and functional activation.** Several positive associations were observed for functional connectivity networks extracted by applying independent component analysis to decompose the resting-state fMRI data of the cohort into 42 independent components. Based on their spatiotemporal resemblance to well-known resting-state networks[28,29], we selected 9 components for further analysis. These components represented the medial and lateral visual, default-mode, dorsal attention, right attention, left attention, salience, sensorimotor and temporal (auditory) networks (Fig. 2A). After correcting for multiple comparisons, we observed a significant positive

association between global memory capacity and functional connectivity within these networks (Fig. 2A): the bilateral occipital cortex for medial ($p < 0.036$) and lateral visual ($p < 0.013$) networks, the bilateral precentral and superior frontal gyrus for the sensorimotor network ($p < 0.001$), the right dorsolateral prefrontal cortex for the right attention network ($p < 0.034$), the right intraparietal sulcus for the dorsal attention network ($p < 0.009$), and the right superior temporal sulcus for the auditory network ($p < 0.036$).

In contrast to the voxel-wise analyses of gray matter volume, which did not show any significant associations with global or regional memory capacity, we found widespread positive correlations between FA values and global memory capacity ($p < 0.05$, corrected for multiple comparisons). These associations were strongest in the bilateral superior longitudinal fascicles connecting the parieto-occipital and frontal cortices, but were also present in the genu and splenium of the corpus callosum, the optic radiations, the right inferior longitudinal tract, uncinate fascicles, and the right corticospinal tract (Fig. 2B).

Finally, we examined whether the global memory capacity was associated with the signal intensity of the locus coeruleus assessed on magnetization transfer imaging. The locus coeruleus is a small nucleus in the brainstem that sends connections to many cortical and subcortical regions, playing a key role in several cognitive and behavioral functions[30,31]. Previous studies have shown that its integrity changes with age, which could affect the brain connectivity patterns and lead to cognitive decline over the course of aging[31,32]. Using linear models, we tested whether the global memory capacity could predict locus coeruleus intensity, which is a measure of its structural integrity[31], in the entire cohort. These analyses did not show any significant relationships. However, when using regional memory capacity as a predictor of the locus coeruleus intensity, we found several significant interactions between age and regional memory capacity values in frontal, temporal, occipital, parietal, and subcortical regions after correcting for multiple comparisons (Fig. 2C, Supplementary Methods 9 and Supplementary Fig. 9), such that higher memory capacity values were associated with smaller decline in locus coeruleus intensity with age.

**Higher global and regional memory capacities are associated with better cognition.** To investigate whether global and regional memory capacity were associated with cognitive domains, such as global cognition, memory, executive functions, visuospatial abilities and psychomotor speed beyond the effects of age, sex and education, we applied partial least squares regressions. We found that regional memory capacity in the middle and inferior occipital gyri, lingual gyrus, fusiform gyrus, Heschl's gyrus, postcentral gyrus, supplementary motor area, gyrus rectus, thalamus, and putamen were all significant predictors of global cognition (Fig. 3A). Memory was best explained by the global memory capacity, as well as by the regional memory capacity in the middle occipital gyrus, gyrus rectus, fusiform gyrus, and inferior parietal gyrus (Fig. 3B). Visual short-term memory, a test measuring both visuospatial abilities and memory, was explained by the gyrus rectus, inferior frontal gyrus (pars orbitalis), superior frontal gyrus, lingual gyrus, and paracentral lobule (Fig. 3C). The inferior frontal gyrus (pars orbitalis), lingual gyrus, superior parietal gyrus, medial orbital gyrus, and putamen were all associated with executive functions (Fig. 3D). Finally, psychomotor speed was best explained by the inferior frontal gyrus (pars orbitalis), postcentral gyrus, and middle occipital gyrus (Fig. 3E). Further details are shown in Supplementary Methods 10 and Supplementary Fig. 10.

**Replication in an independent cohort**
To assess the generalizability of our findings, we evaluated the association between computational memory capacity and aging in an

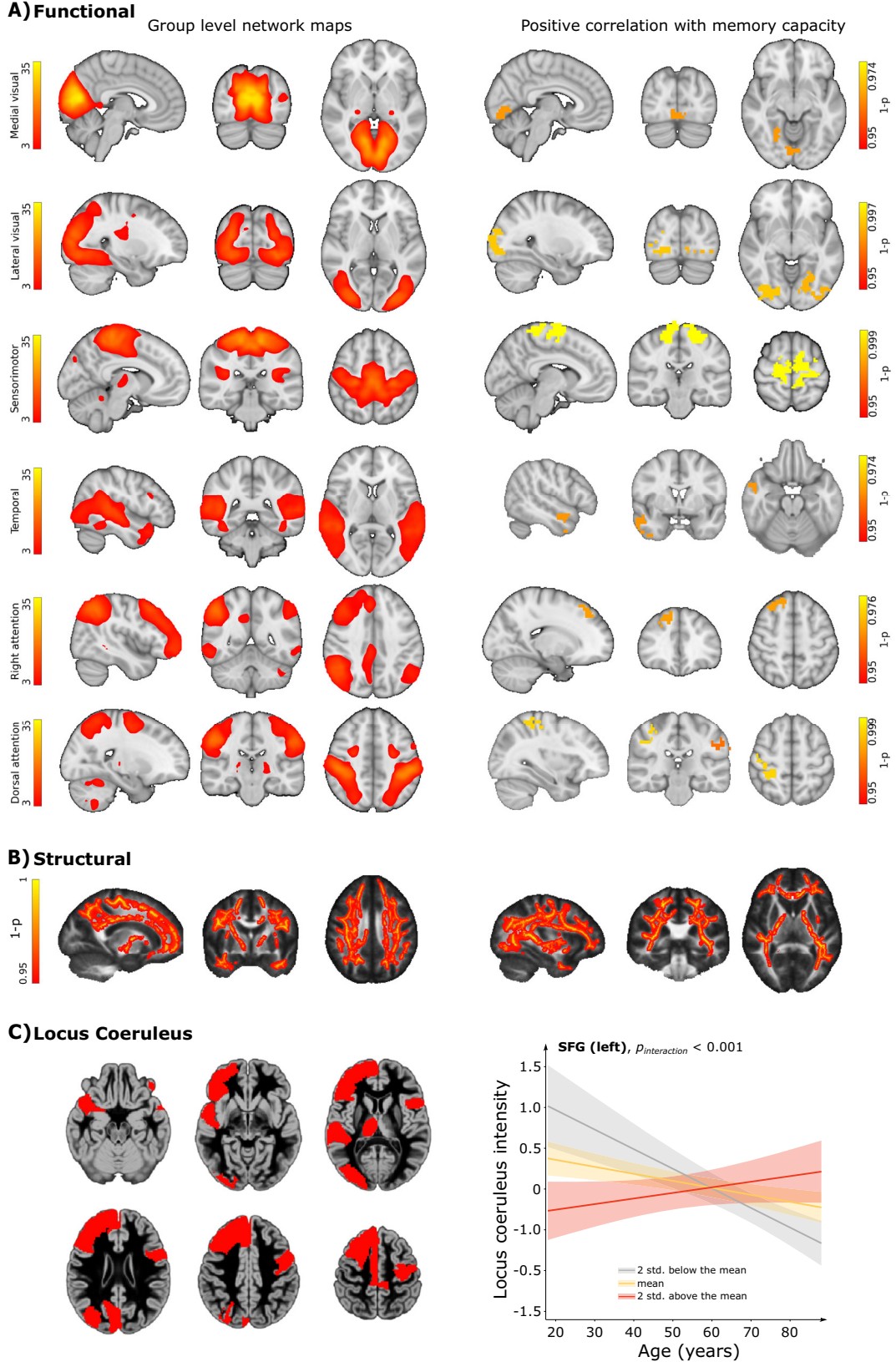

independent cohort, the Leipzig Cohort for Mind-Body-Emotion Interactions[33] (LEMON) study of 154 young and 72 old clinically normal individuals. Similar to our previous results, the global memory capacity was lower in old individuals across the complete density range (Fig. 4A), with larger differences specifically observed for high densities (Fig. 4B, Supplementary Methods 11, and Supplementary

Fig. 11). To evaluate the classification performance of global and regional memory capacity in discriminating young from old individuals, we used a deep-learning model[27]. We evaluated the performance of this model using the receiver operator characteristic (ROC) curve, which plots the true positive rate versus the false positive rate as a function of the threshold for classification. We

**Fig. 2 | Association of memory capacity with measures of structural integrity, functional activation and locus coeruleus intensity. A** Spatial maps of the ICA components corresponding to (from top to bottom) medial visual, lateral visual, sensorimotor, temporal, right attention, and dorsal attention networks (left). The colored areas in the right panel showed significant positive association between global memory capacity and functional connectivity within the corresponding networks. Only networks that showed significant associations after correcting for multiple comparisons are shown overlaid on the MNI152 structural template[93–95]. **B** Visualization of white matter tracts that showed positive association between global memory capacity and FA values. The underlying FA template was created from the average of MNI-registered FA volumes from 20 representative Cam-CAN subjects under 36 years of age. We also used the tbss_fill command in FSL on the FA results, which thickens the thresholded stats image and allows for better visualization of the results. **C** Regions that had significant interaction between regional memory capacity and age in a linear model where the intensity of the locus coeruleus was used as a dependent variable. The average gray matter probability map (MNI152 template)[93–96] shows only the regions with significant interactions after correcting for multiple comparisons (FDR < 0.05) in red. Plot of locus coeruleus intensity as a function of age for superior frontal gyrus, for different levels of memory capacity measured across the sample (level 1: mean memory capacity, yellow; level 2: 2 standard deviations above the mean, gray; level 3: 2 standard deviations below the mean, red). The plotted values are the z-scores of the corresponding raw locus coeruleus intensity values, shaded areas indicate 95% CI for predictions. Source data for Fig. 2 is provided as a Source Data file.

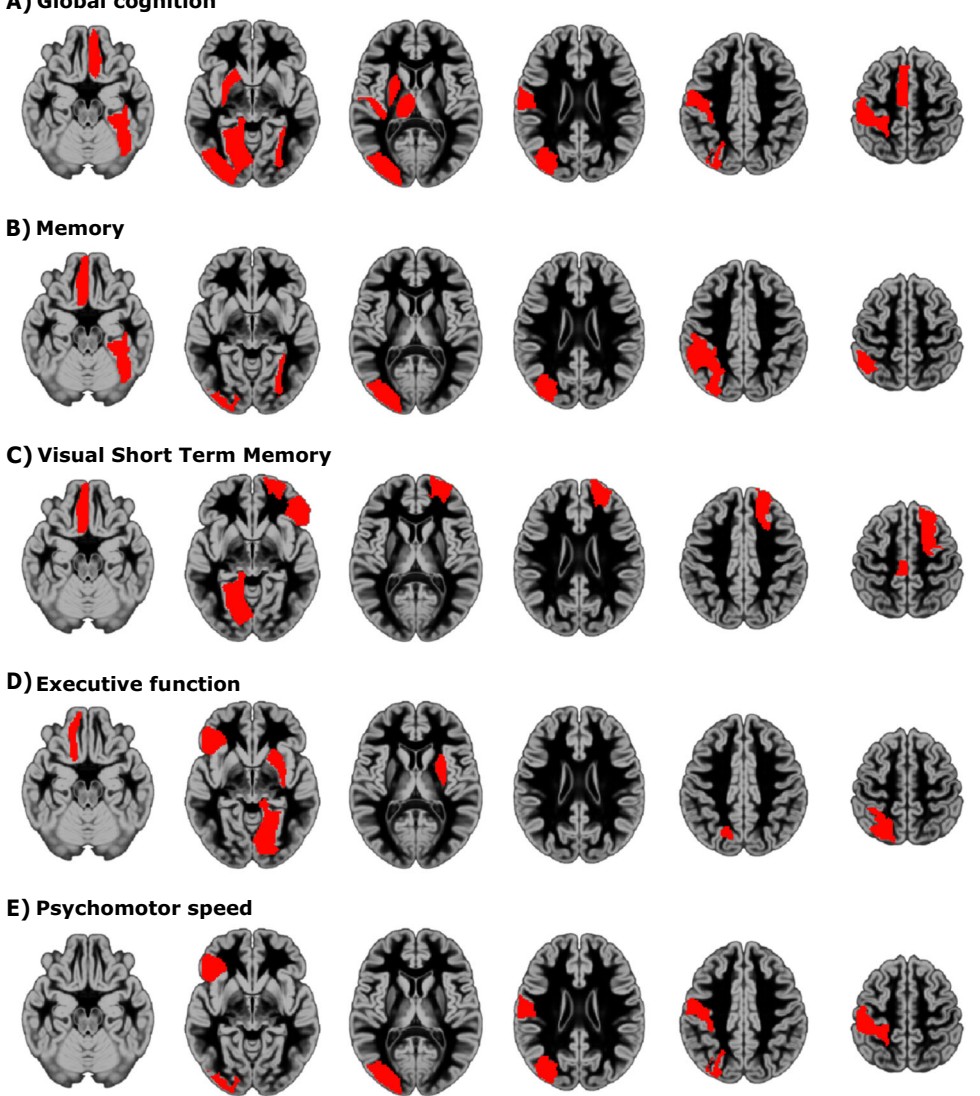

**A) Global cognition**

**B) Memory**

**C) Visual Short Term Memory**

**D) Executive function**

**E) Psychomotor speed**

**Fig. 3 | Regional memory capacity and cognitive performanice.** Visualization of regions whose regional memory capacity was a significant predictor of the individual performance on cognitive tests that measured: **A** global cognition, **B** memory, **C** visual short-term memory, **D** executive function, and **E** psychomotor speed. The significant regions are shown in red, overlayed on an average gray matter probability map (MNI152 template)[93–96]. Source data for Fig. 3 is provided as a Source Data file.

observed that, across 10 cross-validations, the classifier showed a high discriminative accuracy with an AUC of 0.908 ± 0.054 (Fig. 4C), being able to correctly classify 80.3% of the old and 92.7% of the young individuals (Fig. 4D). These findings, in addition to our earlier results, support the important role played by memory capacity in predicting age-related changes.

## Discussion

Memory decline is an important aspect of aging that has led many studies to investigate its underlying causes and potential remedies[34]. Memory has been commonly evaluated in clinical settings using cognitive tests that involve memorizing and retrieving information[35]. Here, instead, we explore memory from a different perspective, as the

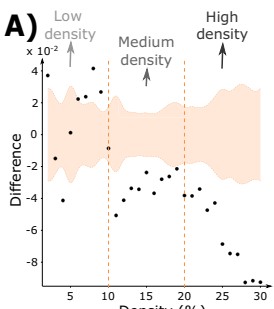
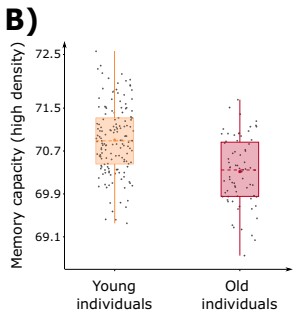
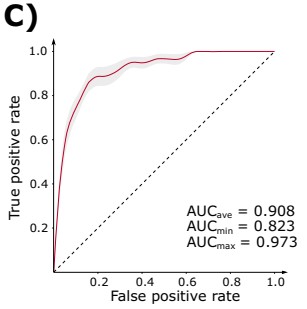
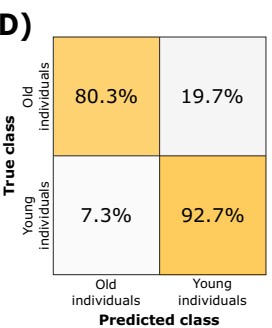

**Fig. 4 | Association between memory capacity and age in LEMON cohort.**
**A** Differences in global memory capacity between older ($n = 72$) and younger ($n = 154$) individuals. The black dots represent differences as a function of network density, while the 95% CI are shown in orange. **B** Boxplots of the global memory capacity at high density values for young ($n = 154$, orange) and old ($n = 72$, red) subjects. The boxplots show the median (center line) and the mean (colored center circles) of each group, the box boundaries correspond to the 25$^{th}$ and 75$^{th}$ percentiles of the sample with whiskers indicating non-outlier data. **C** The receiver operator characteristic (ROC) curve obtained for classification of young vs. old individuals using network and regional memory capacity at high densities (average AUC of 0.908). The red line denotes the mean curve over 10 cross validations and the gray shaded area denotes the standard error of the mean at each point. **D** Confusion matrix for the binary classifier on the test set, showing the percentage correct/wrong classification of old and young individuals across 10 cross-validations, normalized considering each row of the matrix. Source data for Fig. 4 is provided as a Source Data file.

computational capacity of an anatomical brain network to remember or reproduce time-dependent external signals. We achieved this by interpreting the anatomical network as a reservoir computer, a machine-learning concept[10] that permits us to analyze how information flows between regions in the anatomical network, quantifying this ability through the memory capacity measure. The memory capacity declines with aging, allowing us to classify younger vs. older adults with high accuracy and, even, to predict individual ages. Furthermore, beyond the effects of age, we show that memory capacity is associated with white matter, locus coeruleus integrity, and resting-state functional activation signals, therefore relating the computation capability with empirically obtained measures of brain structure and function. Finally, we demonstrate that the computational memory capacity is associated with individual cognitive performance in several domains, including memory. Our findings, summarized in Fig. 5, shed light on the computational measures of brain function that may become important non-invasive imaging biomarkers of aging and provide mechanistic insights into brain network memory.

Previous studies have shown that reservoir computing can offer biologically relevant insights into the performance of anatomical networks on simulated cognitive tasks[11,13]. However, they have left unanswered the question regarding whether the computational memory capacity calculated within this framework has any clinical relevance and whether it is associated with the individual performance in different cognitive tests. By showing that the global memory capacity of the anatomical network decreases with aging, especially at high network densities, our findings suggest that the decline in memory capacity with aging is driven by the weaker connections of the brain network, which may be more vulnerable to aging. To confirm the presence of specific aging effects affecting these connections, we assessed how the memory capacity of different regions changed with age. We observed that the regions that featured the strongest memory capacity decline belonged to the frontal and parietal cortex, in line with earlier studies reporting that frontal and parietal areas are vulnerable to aging[6,36]. Specifically, a strong decrease was observed in the anterior cingulate cortex, which is known to play an important role in executive functions, such as decision-making, error detection, and emotion regulation[37,38]. Furthermore, memory capacity also decreased in medial parietal areas, including the precuneus. The precuneus is a central hub in the brain, being able to integrate information from many areas, which provides it with an ability to perform a wide range of functions including visuo-spatial imagery, episodic memory retrieval, and cognitive control[39]. A decrease of memory capacity in these

regions indicates a reduced capacity to integrate information from multiple brain regions, which could lead to a decline in complex cognitive functions, including intelligence[40]. Moreover, the simultaneous decrease of memory capacity in both of these regions could also reflect reductions in anterior-to-posterior connectivity observed in the default mode network during aging[41].

Interestingly, we also observed that the memory capacity increased with age in the parahippocampal and Heschl's gyri. The parahippocampal gyrus is part of the parahippocampal cortex, which is a component of a large network that connects temporal, parietal, and frontal regions[42]. It has been implicated in various cognitive functions, but most consistently in episodic memory and visuospatial processing[42]. Considering the highly interconnected nature of the parahippocampal cortex, the observed memory capacity increases in the parahippocampal gyrus could represent an attempt to maintain overall network performance by over-recruiting hetero-modal regions in order to compensate for the loss of function in other regions that are part of this network. This tendency to over-recruit functional systems at the expense of their functional specialization has been previously observed in older individuals in temporal areas, and can be partly explained by the age-related decline in white matter integrity[23,24,43,44]. Furthermore, the increase of memory capacity in Heschl's gyrus is in line with previous studies showing that older adults need to recruit additional resources to cope with an increased hearing effort[25,26], which could manifest as increased memory capacity in this region.

An important finding from our study is that the regional and global memory capacity accurately predicts the individual age in a lifespan cohort of healthy individuals using a deep learning model, which reached a high correlation between true and predicted age of up to 0.78. To assess the generalizability of our findings, we replicated them in an independent cohort consisting of two groups of young and old individuals. We found that memory capacity was able to obtain high classification accuracy with an average AUC of 0.91. Furthermore, across 10 cross-validations, the model was able to classify correctly 93% of the young and 80% of the old individuals. These findings indicate that the memory capacity variability is higher in older individuals than in younger ones, as reflected in the lower classification accuracy for old individuals, which is consistent with previous findings showing a higher variability in cognitive performance in old individuals[45].

In order to directly examine the relationship between cognitive performance and memory capacity at the level of the whole brain and of individual brain regions beyond the effects of age, sex and

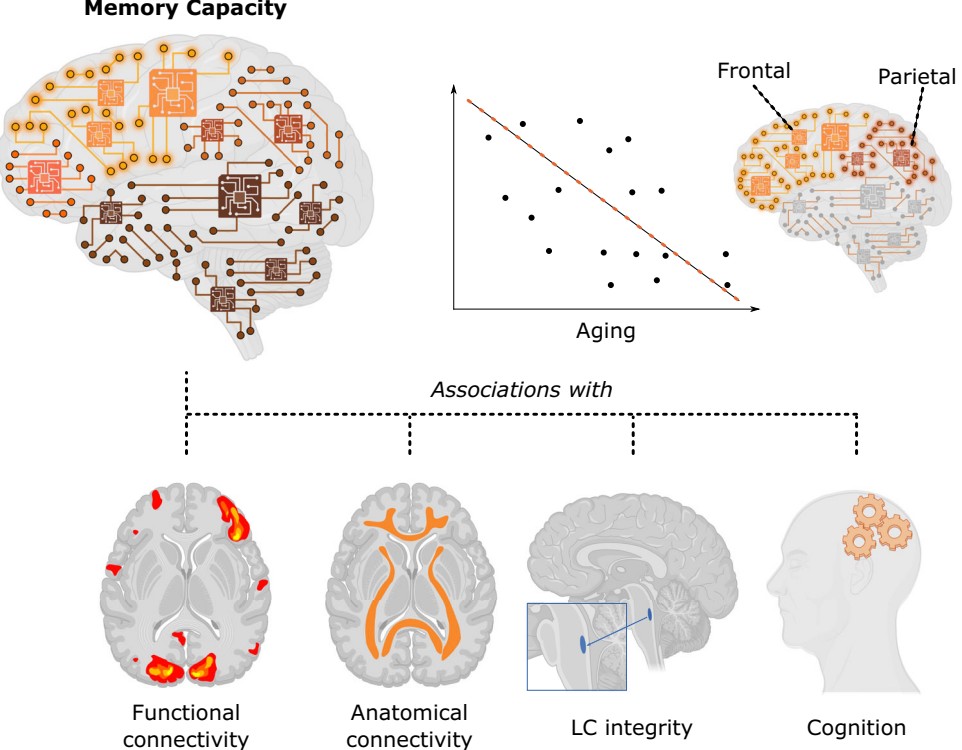

**Fig. 5 | Overview of the main findings.** The computational memory capacity decreases with aging (Created in BioRender. Mijalkov, M. 2025 https://BioRender.com/l54p851), with the strongest effects being observed in frontal and parietal regions (Created in BioRender. Mijalkov, M. 2025 https://BioRender.com/p48p972). Furthermore, the memory capacity showed significant associations with brain function (Created in BioRender. Mijalkov, M. 2025 https://BioRender.com/ n07i150), structure (Created in BioRender. Mijalkov, M. 2025 https://BioRender.com/c23d438), integrity of locus coeruleus (Created in BioRender. Mijalkov, M. (2025) https://BioRender.com/h54q798; Created in BioRender. Mijalkov, M. 2025 https://BioRender.com/r26i595), as well as cognitive performance in multiple domains (Created in BioRender. Mijalkov, M. 2025 https://BioRender.com/ e65y790).

education, we applied a partial least squares analysis. We measured memory function by the face recognition test, since age-related decline in face recognition and memory are closely related[17]. We found that memory function was associated with global memory capacity. Furthermore, it was associated with the regional memory capacity of occipital regions, gyrus rectus, fusiform gyrus, and inferior parietal gyrus. These regions are involved in visual and emotional processing, memory, and multisensory integration of facial stimuli[46–48]. In fact, the regional memory capacity of occipital regions were significant predictors of scores across all tasks, reflecting the visual component of each test[17]. We used the hotel task to assess executive function, which involves complex planning, multitasking and adaptation. We found that executive function was related to regional memory capacity in regions involved in different aspects of executive function, including the superior parietal gyrus, which is involved in spatial attention, working memory, and goal updating[49,50]. Psychomotor speed, measured by the "choice" response time task, was explained by the regional memory capacity of the inferior frontal and the postcentral gyrus, which are involved in language, inhibition, attentional control, somatosensory processing, proprioception, and sensory-motor integration[51,52]. Visual short-term memory was associated with the regional memory capacity of the gyrus rectus, paracentral lobule, and inferior and superior frontal gyri. Although these regions have not been directly associated with visual short-term memory, this finding could reflect the involvement of other domain-general brain regions, as well as the activation of the frontal-parietal network while performing visual short-term memory tasks[53,54]. Finally, we measured global cognition using the proverbs test, which was explained by multiple regions that are all involved in different aspects of cognition, such as visual processing, sensory-motor integration, language,

memory, and executive function. Our results suggest that cognitive performance is linked to the computational ability both at the global and regional levels, and that different aspects of cognition rely on different brain regions that enable the perception, representation, manipulation, and integration of information across domains and modalities[25,46,47,52,55,56].

The global memory capacity was also associated with higher FA values in several white-matter tracts that have been shown to mediate complex cognitive tasks, which can be interpreted as preserved integrity of white matter fibers in these bundles[57]. Higher global memory capacity was related to preserved integrity in the bilateral superior longitudinal fascicles connecting parieto-occipital with frontal regions, which plays an important role in language, attention, memory, and emotions[58,59]. In addition, we found that memory capacity was associated with the integrity of the genu and splenium of the corpus callosum. The genu and splenium play an important role in integrating information from different brain regions by making extensive connections with frontal regions (the genu) and occipital-parietal and temporal cortex (the splenium), being essential for interhemispheric communication across different brain regions[60].

We also investigated the relationship between global memory capacity and voxel-wise functional activation within resting-state functional MRI networks. We found that increased memory capacity was associated with increased functional activation in regions belonging to the sensorimotor, frontoparietal, dorsal attention, auditory, as well as medial and lateral visual networks[29,61]. This shows that the computational capability of the network is not specific to any particular function, but rather reflects more general widespread functional connectivity patterns. Therefore, the link between network memory capacity and functional activation, which is an emergent

**Table 1 | Characteristics of the individuals from the Cam-CAN cohort**

|  | Cam-CAN (n = 636) |
|---|---|
| Age (years) | 18–88 (55) |
| Sex (M/F) | 315/321 |
| Education (years) | 14–36 (21) |
| Proverb comprehension | 0–6 (5) |
| Familiar faces recognition | 5–30 (26) |
| Visual short-term memory | 0.23–1.03 (0.45) |
| Hotel task | 24.25–960 (267.93) |
| "Choice" response time | 0.29–1.21 (0.77) |

The range of values for each row is followed by the median in parenthesis, except for sex.

property of the anatomical connectome that can be measured empirically, highlights the importance of computational capacity as an inherent property of brain networks.

Finally, to gain insights into how memory capacity is related to other important measures of brain integrity and function, we assessed its association with the signal intensity of the locus coeruleus. The locus coeruleus is a small subcortical brain structure that modulates various cognitive and behavioral functions, including executive functions, attention, and memory, through the release of norepinephrine, a key neuromodulator in the brain[62]. It is connected to almost all brain regions, shaping and coordinating widespread neural activity[30]. Aging has been previously associated with changes in locus coeruleus integrity, which can be measured as signal hypointensities in locus coeruleus contrast magnetic resonance sequences such as magnetization transfer imaging[30,32]. These changes have been shown to be associated with changes in cognitive performance, especially in older adults[30,31]. Here, we examined whether memory capacity has a role in modulating the observed changes in locus coeruleus integrity with age. We found an interaction between regional memory capacity values and age in the predictions of locus coeruleus intensity, such that high memory capacity values were associated with slower decline in locus coeruleus intensity with increasing age. This effect was observed for various frontal, parietal, occipital, and temporal regions, as well as for the thalamus and putamen. Our findings are consistent with the widespread projections of the locus coeruleus throughout the brain, suggesting that the improved noradrenergic transmission, as facilitated by the higher locus coeruleus integrity, allows for a higher computational capability of the brain network.

Our study has some limitations that should be considered when interpreting our results. Specifically, we used cross-sectional data from two independent cohorts to test the age-related associations between memory capacity and different empirical measures. This design did not allow us to test the causal relationships between memory capacity and these parameters or whether memory capacity could predict cognitive trajectories over time. Since longitudinal studies could offer additional insights about aging trajectories compared to cross-sectional studies[63], future studies are needed to test the predictive value of memory capacity. From a methodological point of view, since our main objective was to show that memory capacity is a marker for aging, we used a homogeneous activation function for all brain regions to allow for easier interpretation of our results. Testing our findings for region-dependent activation functions was beyond the scope of the current work; however, it could allow building more complex models to investigate the role of each individual region in the computational capabilities of the brain network.

Despite these limitations, we have demonstrated that the computational capability of the brain anatomical network, measured by its memory capacity, is an important imaging biomarker for aging that can predict individual age with high accuracy.

Moreover, we show that memory capacity, which has traditionally been considered as an abstract computational property of a network, is associated with brain structure, function, and cognitive performance, offering new insights into how the brains of older individuals decline. This study opens new avenues for applying reservoir computing to study the properties of brain networks, by establishing the link between computational capacity and the empirically measured parameters in clinical settings.

## Methods

### Participants

A total of 636 healthy participants from the Cam-CAN database[17] (Table 1) aged from 18 to 88 years were included in this study. All subjects underwent a brain imaging protocol that included diffusion-weighted imaging, resting-state functional MRI, and magnetization transfer imaging. In addition, a subsample of 462 participants completed a comprehensive assessment of cognitive functions, specifically, global cognition (proverb comprehension task[17,64]), memory (face recognition: familiar faces task[17,65]), visual short-term memory (visual short-term memory task[17,66]), executive function (hotel task[17,67]), and psychomotor speed ("choice" response time task[17]). To assess the reproducibility of the memory capacity results, we repeated the main analyses using the data of 226 participants from the LEMON cohort[33] who also underwent diffusion-weighted imaging and were divided into 154 young (20–40 years, 109 male, 45 female) and 72 old (55–80 years, 36 male, 36 female) individuals. Both cohorts obtained written informed consent from all participants. The Cam-CAN cohort study was conducted in compliance with the Helsinki Declaration, and has been approved by Cambridgeshire 2 Research Ethics Committee (reference number: 10/H0308/50). The LEMON cohort study was also carried out in accordance with the Helsinki Declaration and the study protocol was approved by the ethics committee at the medical faculty of the University of Leipzig (reference number 154/13-ff). We did not use any additional data or re-contact the participants and signed data use agreements to analyze the data.

### Reservoir computing

We employed reservoir computing to evaluate the capacity of the anatomical networks to retain the temporal features of an input signal. Reservoir computers have three components: an input layer, a reservoir, and a readout layer. The readout layer used linear units, while reservoir nodes were tanh units. Here, a random input time series, $u(t)$, was generated for 20,000 time steps $t$. At each time step, the signal $u(t)$ was independently sampled from a uniform distribution in the interval (0, 1). Following the standard approach for reservoir computing, we fed the same time series to all reservoir nodes[10,15], multiplying it by an input vector $\mathbf{W}_{in}$ of size $N_r \times 1$, where $N_r$ denotes the number of nodes in the reservoir, each associated to a different anatomical region. We randomly assigned the components of $\mathbf{W}_{in}$ from the standard normal distribution.

Reservoir internal connections $\mathbf{W}$, of size $N_r \times N_r$, were derived from the individual diffusion-weighted imaging scans. To ensure a meaningful comparison across different individuals, we kept only a proportion of the network's strongest connections (1–30% density in steps of 1%, while keeping the connectivity strength of individual connections), and compared the reservoir performance across this range of connection densities. We scaled the weights of all connectivity matrices by dividing them to the matrix largest singular value (spectral normalization) to ensure stable reservoir dynamics and better performance[16], however, we found that the performance of the model was robust against the variation of this parameter. Further details regarding the choice of the normalization parameter are presented in Supplementary Methods 12, Supplementary Table 2, and Supplementary Fig. 12.

The reservoir dynamics was governed by the following model:

$$\mathbf{r}(t+1) = \tanh\big(\mathbf{W}_{\text{in}}u(t) + \mathbf{W}\mathbf{r}(t)\big), \qquad (1)$$

where $\mathbf{r}(t)$ is a vector of size $N_r \times 1$ that represents the reservoir states at time $t$ and the tanh activation function is applied element wise. As $\mathbf{W}$ represents the brain connectome derived from diffusion-weighted images, it is a symmetric matrix containing only positive connections. In summary, the reservoir states at time $t+1$ are nonlinearly influenced by the input, and the weighted sum of the previous reservoir states. We initialized the reservoir states with $\mathbf{r}(t) = 0$, resulting in an initial transient activity of the system that was not related to the input. Therefore, we discarded the initial 5% time steps of the input, and the recorded reservoir states, in all cases.

The output prediction from the reservoir can be obtained as:

$$\hat{\mathbf{y}} = \mathbf{W}_{\text{out}}\mathbf{R}, \qquad (2)$$

where $\hat{\mathbf{y}} = \big[\hat{y}(t), \hat{y}(t-1)\ldots \hat{y}(1)\big]^{\mathrm{T}}$ is a vector of temporally concatenated output values with dimension of $n_{signal} \times 1$, and $n_{signal}$ is the length of the time activation course of each node. $\mathbf{R}$ is a $N_r \times n_{signal}$ matrix, where each column contains the activation states $\mathbf{r}(t)$ for all nodes at different time steps. To calculate regional memory capacity, $\mathbf{R}$ is instead a $1 \times n_{signal}$ vector, and the output prediction is computed for each node independently.

Training the model entails calculating $\mathbf{W}_{\text{out}}$, which is done in the following manner[10]:

$$\mathbf{W}_{\text{out}} = \mathbf{y}_\tau\, \mathbf{R}^{\mathrm{T}}\mathbf{X}_{\mathrm{R}}. \qquad (3)$$

In the above, $\mathbf{R}^{\mathrm{T}}$ denotes the transpose of $\mathbf{R}$, $\mathbf{X}_{\mathrm{R}}$ represents the Moore-Penrose pseudoinverse[10] of $\mathbf{R}\mathbf{R}^{\mathrm{T}}$, and $\mathbf{y}_\tau$ is the array of delayed input signals as described in the following section.

## Memory capacity

To compute the linear memory capacity of the reservoir, the network is trained to reproduce delayed representations of the input signal[15]. Specifically, $y_\tau(t) = u(t - \tau)$, where $\tau$ represents the amount of time steps $y_\tau(t)$ is delayed with respect to the input signal. Therefore, for each time delay $\tau$, the reservoir is independently trained to predict $y_\tau(t)$. The memory capacity of the reservoir is then calculated by evaluating the Pearson's correlation coefficient, $\rho$, between the reservoir output, $\hat{y}_\tau(t)$, and the delayed input signal, $y_\tau(t)$:

$$MC = \sum_{\tau=0}^{\infty} \rho\big(\hat{y}_\tau(t), y_\tau(t)\big). \qquad (4)$$

In this work, we evaluated the memory capacity in the range $\tau = 6$ to $\tau = 35$. We used this interval for $\tau$ since at lower $\tau$ values the network was able to memorize the input signals almost perfectly regardless of age, while it failed to do so for values of $\tau$ larger than 35. The memory capacity values were calculated as an average from 10 trials, where in each trial we used different realization of the random input signals and $\mathbf{W}_{\text{in}}$.

## Age prediction

The deep-learning method used for the prediction of age was a multilayer perceptron, a widely used neural network architecture that consists of an input and output layers connected nonlinearly by hidden layers and trained using error backpropagation[27,68]. Here, we used a neural network consisting of four hidden layers with 256, 512, 512, and 256 nodes (Fig. 1J). To prevent model overfitting, we applied dropout regularizations with a 10% rate (1 out of 10 inputs were randomly dropped during training) between the hidden layers. The densely connected output layer was activated with a linear (in the case of regression predictions) or sigmoid (in the case of classification predictions) functions. The model performance was assessed using 10-fold cross validation. For each fold of the cross validation, the input data was divided into training, validation, and test datasets (80%, 10%, and 10%) and normalized (between 0 and 1). The multilayer perceptron was trained to minimize the mean absolute error (for the regression task or age predictions in the Cam-CAN cohort) or binary cross entropy loss (for the binary classification task or the young versus old subjects classification in the LEMON cohort), using the Adam optimizer with a learning rate of $5 \times 10^{-4}$ and batch size of 64 during 100 epochs. For each fold, we chose the final model with the lowest validation loss and evaluated its performance on the previously separated test data.

## Image acquisition

In the Cam-CAN cohort, the images were collected at a single site using a 3 T Siemens TIM Trio scanner with a 32-channel head coil. The diffusion weighted data was acquired with a twice-refocused-spin-echo sequence, with 30 gradient directions for b-values 1000 and $2000\,\mathrm{s\,mm^{-2}}$, while three images were acquired using a b-value of 0. The following parameters were used: $TE = 104\,\mathrm{ms}$, $TR = 9100\,\mathrm{ms}$, field of view $(FOV) = 192\,\mathrm{mm} \times 192\,\mathrm{mm}$, voxel size $= 2 \times 2 \times 2\,\mathrm{mm^3}$, 66 axial slices, GRAPPA acceleration factor 2. The T1-weighted scans were acquired using a MPRAGE sequence with parameters: $TR = 2250\,\mathrm{ms}$, $TE = 2.99\,\mathrm{ms}$, $FOV = 256\,\mathrm{mm} \times 240\,\mathrm{mm} \times 192\,\mathrm{mm}$, voxel size $= 1 \times 1 \times 1\,\mathrm{mm^3}$, GRAPPA factor 2. The resting-state functional MRI images were acquired with an echo planar imaging (EPI) sequence: $TR = 1970\,\mathrm{ms}$, $TE = 30\,\mathrm{ms}$, $FOV = 192\,\mathrm{mm} \times 192\,\mathrm{mm}$, voxel size $= 3 \times 3 \times 4.44\,\mathrm{mm^3}$, 261 volumes, each containing 32 axial slices of thickness 3.7 mm. Magnetization Transfer Ratio (MTR) images were obtained from two MT-prepared Spoiled Gradient (SPGR) sequences: $TR = 30\,\mathrm{ms}$ (or $TR = 50\,\mathrm{ms}$ if SAR surpassed limits), $TE = 5\,\mathrm{ms}$, $FOV = 192\,\mathrm{mm} \times 192\,\mathrm{mm}$, voxel size $= 1.5 \times 1.5 \times 1.5\,\mathrm{mm^3}$, bandwidth $= 190\,\mathrm{Hz/px}$. The applied pulse was a Gaussian RF pulse with an offset frequency of 1950 Hz and duration 9984 μs (bandwidth = 375 Hz, flip angle = 500°). Regarding the LEMON cohort, the diffusion weighted data were acquired with multi-band accelerated sequence and an in-plane GRAPPA (acceleration factor 2). The parameters were as follows: 88 axial slices, voxel size $= 1.7 \times 1.7 \times 1.7\,\mathrm{mm^3}$, 60 gradient directions, b-value of $1000\,\mathrm{s\,mm^{-2}}$, $TR = 7000\,\mathrm{ms}$, $TE = 80\,\mathrm{ms}$, $FA = 90°$, $FOV = 220\,\mathrm{mm}$, in addition to 7 images acquired using a b-value of 0. More details are available in[17,33].

## Preprocessing of diffusion-weighted scans

The DWI images from both cohorts were preprocessed simultaneously with the same pipeline in order to ensure consistency. The imaging data was preprocessed using FSL[69] (v6.0.7, https://fsl.fmrib.ox.ac.uk/fsl/fslwiki). It was corrected for motion and eddy current using EDDY[70] and skull stripped using BET[71], both of which are distributed with FSL, and then diffusion tensors were fit to the data using dtifit. Using the fitted tensors, we derived FA maps for each individual.

## Preprocessing of functional MRI scans

We preprocessed the functional scans from the Cam-CAN dataset using a standard pipeline in fMRIPrep[72] (v20.2.4, https://fmriprep.org/en/stable/). We first removed the first two volumes to allow for steady state magnetization, followed by correcting the images for motion and slice timing effects. The functional images were then skull-stripped and, using a two-stage registration approach with Freesurfer[73] (v7.4.1) and ANTs[74] (v2.4.4), co-registered to a standard 2mm resolution MNI152 template space. The resulting images additionally underwent motion correction using the Friston-24 head motion model[75] and nuisance regression to remove confounding signals from the white matter and cerebrospinal fluid. The resulting volumes were spatially smoothed using an isotropic Gaussian kernel with 6mm FWHM.

## Delineation of the locus coeruleus and obtaining locus coeruleus contrast intensity values

To derive a measure of locus coeruleus (LC) integrity for each participant, we used the magnetization transfer-weighted (MTW) images from the Cam-CAN cohort, which were shown to reflect age-related differences in LC integrity[76]. After skull stripping the MTW images using FSL BET[71], we used bias field correction and nonlinear registration as implemented in ANTs[74] to transform them to MNI space. We extracted average LC signal intensity values from a group consensus mask[76]. To calculate LC contrast, we first delineated a reference region in the pons[77] and calculated the LC contrast as $(S_{LC} - S_{ref})/S_{ref}$, where $S_{LC}$ denotes the mean signal from the LC and $S_{ref}$ is the mean signal from the pontine reference region.

## Calculation of whole-brain anatomical connectivity networks

We calculated the connectivity networks using a deterministic tractography pipeline[78], implemented in DSI Studio (v2023.07.08, http://dsi-studio.labsolver.org), by performing fiber tracking on the individual FA maps. Anatomical connectivity was defined as the number of streamlines connecting each pair of the regions defined by the automated anatomical labeling atlas[18]. We thresholded the resulting connectivity networks by keeping only the connections with a coefficient of variation equal to or below the 10th percentile across the sample population[19]. We analyzed connectivity networks between 94 regions, excluding all regions within the cerebellum and upper cervical spinal cord since they were not fully covered by the fMRI acquisition used in the Cam-CAN cohort. To evaluate the robustness of our findings against different preprocessing pipelines, we replicated our results using probabilistic tractography to build the anatomical connectivity networks (Supplementary Methods 13 and Supplementary Fig. 13).

## Tract-based spatial statistics

In order to assess how memory capacity correlates with white matter integrity, we conducted a regression analysis using tract-based spatial statistics (TBSS) as implemented in FSL[79]. First, FA volumes of each participant were re-aligned to the FMRIB template in MNI space using nonlinear registration with FSL FNIRT. Then, a mean FA volume was derived by calculating the average of all participants' FA images. The mean FA volume was thresholded at FA value of 0.2 and skeletonized to represent the center of the white matter tracts. The template-aligned FA volumes of all participants were then projected to this mean FA skeleton. The relationship between global memory capacity and diffusion parameters was assessed using a general linear model (GLM) approach, with age and biological sex included as confound regressors in the model. The GLM was fitted voxel-wise and statistical inference was performed using a non-parametric permutation test as implemented in FSL randomise[80]. The spatial relationship between voxels was taken into account by carrying out threshold-free cluster enhancement (TFCE) with 2D optimization[81]. Correction for multiple comparisons was performed by controlling the family wise error rate ($p < 0.05$).

## Group independent component analysis

The spatiotemporal properties of resting-state networks (RSNs) vary with age[61], so we used a reference group of 125 participants under 36 years of age[82] to derive a spatial template for well-known resting-state networks to assess the relationship between global memory capacity and voxel-wise functional connectivity within these networks. Group-average resting-state network maps from the template cohort were obtained using temporal concatenation ICA as implemented in FSL MELODIC[83] (v3.15). The number of components was determined automatically using the Laplacian approximation to the posterior distribution of the model order. Independent component spatial maps were thresholded by fitting a Gaussian-Gamma mixture-model to the Z-transformed intensity value histogram. Components were chosen for further analysis based on their spatiotemporal resemblance of known resting state networks[28,29]. Individual versions of group-level resting-state network maps identified from the group ICA analysis were estimated using dual regression[84]. The relationship between global memory capacity and functional connectivity was assessed voxel-wise for each RSN separately using a GLM-based non-parametric permutation approach similarly to the DTI analysis, with FSL randomize, controlling for age and biological sex. The analysis was constrained to the area under the thresholded group-average network maps.

## Statistical analysis

We assessed the statistical significance of the differences between old and young individuals by nonparametric permutation tests with 10,000 permutations, which were considered significant for a two-tailed test of the null hypothesis at $p < 0.05$. We used area under curve (AUC) analysis to summarize the memory capacity of each individual across low, medium, and high densities. This approach considers the complete density range, and therefore, is less sensitive to the thresholding process. We obtained an AUC estimate for each category by numerically integrating memory capacity values over the corresponding density range, yielding 3 memory capacity values per individual. Each of the 3 values were included as dependent values in separate linear models, with age as independent values. All analysis used sex as a covariate. To evaluate the relation between cognition and memory capacity, we used partial least squares analysis. It is a statistical technique that linearly decomposes the predictor and predicted variable matrix into latent variables (LVs), which are linear combinations of the original variables that are optimized so that the covariance between the resulting predictor and predicted matrix components (factors and loadings) is maximal[85]. We fitted a separate model for each cognitive test, which included age, sex, education, and global and regional memory capacity as predictors. The optimal number of LVs was 5 for memory, visual short-term memory, and psychomotor speed scores, while it was 4 for global cognition and executive function. Each variable's contribution to the prediction was quantified using variable importance in the projection (VIP) scores. VIP scores were calculated by summing the partial least squares weights over latent variables (LVs), and weighting them by the variance explained by each LV. Variables were considered to be significant contributors to the prediction if their VIP score was greater than unity[86].

## Reporting summary

Further information on research design is available in the Nature Portfolio Reporting Summary linked to this article.

# Data availability

The data used in this study was obtained from Cambridge Centre for Ageing and Neuroscience (Cam-CAN) cohort[17] (https://www.cam-can.org/). As a replication cohort, we used the Leipzig Mind-Brain-Body Dataset−LEMON[33], which can be accessed at https://openneuro.org/datasets/ds000221/versions/00002. Both cohorts are open-access and require an application for access. Source data are provided with this paper.

# Code availability

The reservoir computing paradigm and memory capacity were calculated using a modified version of BRAPH 2.0[87] using the pipeline "Memory Capacity". BRAPH software can be freely downloaded from http://braph.org/ (https://github.com/braph-software/BRAPH-2), along with detailed user manuals of how to upload and analyze the data. The modified version of BRAPH 2.0 is hosted on https://github.com/braph-software/MemoryCapacity[88]. All deep learning methods were developed in Python (v3.8.10, https://www.python.org/) using the following open-source packages: Pandas[89] (v1.5.3) and Numpy[90] (v1.21.3) for data

handling and Scikit-learn[91] (v1.2.1) and TensorFlow (v2.7.0-gpu, Keras based)[92] for neural network development and evaluation.

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

## Acknowledgements

This work was supported by a Swedish Research Council grant (#2022-01108); Alzheimer Foundation; Brain Foundation (#2022-014); by European Union – NextGenerationEU and the Romanian Government, under National Recovery and Resilience Plan for Romania, contract no. 760250/28.12.2023, cod PNRR-C9-I8-CF109/31.07.2023, through the Romanian Ministry of Research, Innovation and Digitalization, within Component 9, Investment I8 ; Strategic Research Area Neuroscience (StratNeuro); KI Consolidator grant, Center for Medical Innovation (CIMED); Konung Gustaf V:s och Drottning Victorias Stiftelse; Foundation for Geriatric Diseases at Karolinska Institutet; Gamla Tjänarinnor; Stohnes Foundation; Lars Hiertas Memorial Foundation obtained by J.B.P. or M.M. Additionally, H.Z. (Henrik Zetterberg) is a Wallenberg Scholar supported by grants from the Swedish Research Council (#2022-01018 and #2019-02397), the European Union's Horizon Europe research and innovation programme under grant agreement No 101053962, Swedish State Support for Clinical Research (#ALFGBG-71320), the Alzheimer Drug Discovery Foundation (ADDF), USA (#201809-2016862), the AD Strategic Fund and the Alzheimer's Association (#ADSF-21-831376-C, #ADSF-21-831381-C, and #ADSF-21-831377-C), the Bluefield Project, the Olav Thon Foundation, the Erling-Persson Family Foundation, Stiftelsen för Gamla Tjänarinnor, Hjärnfonden, Sweden (#FO2022-0270), the European Union's Horizon 2020 research and innovation programme under the Marie Skłodowska-Curie grant agreement No 860197 (MIRIADE), the European Union Joint Programme—Neurodegenerative Disease Research (JPND2021-00694), the National Institute for Health and Care Research University College London Hospitals Biomedical Research Centre, and the UK Dementia Research Institute at UCL (UKDRI-1003). The data analysis was enabled by resources provided by the National Academic Infrastructure for Supercomputing in Sweden (NAISS), partially funded by the Swedish Research Council through grant agreement no. 2022-06725. Figures 1, 2A, C, and 3 used the MNI152 template: Copyright (C) 1993–2004 Louis Collins, McConnell Brain Imaging Centre, Montreal Neurological Institute, McGill University.

## Author contributions

M.M.: conceptualization, formal analysis, methodology, visualization, writing—original draft, review and editing. LS: methodology. BZG: formal analysis—age prediction. DV: formal analysis and data preprocessing. A.C.G.: visualization, software. E.G.R. and Y.W.C.: software. G.V.: ideation, conceptualization, methodology, writing—review and editing. JBP: conceptualization, methodology, funding, writing—original draft, review and editing. M.M., L.S., B.Z.G., D.V., Z.X., A.C.G., J.S., Y.W.C., H.Z. (Hang Zhao), E.G.R., M.P., S.G.P., M.K., P.S., H.Z. (Henrik Zetterberg), H.J., K.L., D.B., B.M., G.V., J.B.P. reviewed and edited the manuscript.

## Funding

## Competing interests

H.Z. (Henrik Zetterberg) has served at scientific advisory boards and/or as a consultant for Abbvie, Acumen, Alector, Alzinova, ALZPath, Annexon, Apellis, Artery Therapeutics, AZTherapies, Cognito Therapeutics, CogRx, Denali, Eisai, Nervgen, Novo Nordisk, Optoceutics, Passage Bio, Pinteon Therapeutics, Prothena, Red Abbey Labs, reMYND, Roche, Samumed, Siemens Healthineers, Triplet Therapeutics, and Wave, has given lectures in symposia sponsored by Alzecure, Biogen, Cellectricon, Fujirebio, Lilly, and Roche, and is a co-founder of Brain Biomarker Solutions in Gothenburg AB (BBS), which is a part of the GU Ventures Incubator Program (outside submitted work). The other co-authors declare no competing interests.
