## [Transparent Peer Review file · Nature Communications]

Computational memory capacity predicts aging and cognitive decline

Corresponding Author: Dr Mite Mijalkov

Version 0:

Reviewer comments:

Reviewer #1

(Remarks to the Author)

In their manuscript, the authors utilize structured/spatially-embedded reservoir computing (RC) models to explore changes in the computational capacity of the brain's structural network due to aging. By comparing global and regional memory capacity (MC) between "young" and "old" subjects, the authors propose the MC metric, likely a vector of MC values, as a predictor of age. Additionally, they employ advanced machine learning techniques to establish a link between regional MCs and underlying transcriptomic profiles, along with their association with resting state functional activities and scores in five cognitive tasks.

The investigation into the contribution of connectomic features to the memory capacity of a reservoir computing model, as presented in Fig. 1, is intriguing. The results in Fig. 1F-H suggest that increasing connection density leads to a slight increase and subsequent decrease in memory capacity. Speculatively, the low-density regime results in uncontrolled connectedness, forming several small clusters with similar representations within each cluster but distinct representations between clusters. In contrast, the high-density regime, with a more connected network, exhibits less diverse representations, resulting in lower memory capacity. The medium-density range appears to strike a balance by fostering a few weakly connected clusters that guarantee sufficiently diverse representations.

Exploration of brain anatomical changes throughout aging and their computational implications are interesting. However, my enthusiasm for the work wanes due to the incorporation of multiple sub-projects. These projects divert attention from the core themes (structure, memory capacity, and aging) and instead emphasize extracting numerous features from structural, functional, and cognitive datasets. The focus shifts towards establishing correlations between these features and memory capacities. The subsequent listing of concerns is intended to address these deviations and restore clarity to the primary objectives of the study.

- The computation of regional memory capacity, a focal point in this study, raises concerns due to its inherent ambiguity. The results section defines it as "the ability to reproduce the delayed input signal using only the activation time series of single brain regions," while the introduction states it as extending "either across all reservoir nodes (global memory capacity) or across a select subset (regional memory capacity)." A closer look at the shared codes suggests that regional memory capacity evaluates how effectively individual nodes can preserve the memory of the input. Despite comprehending the rationale behind this approach, I remain unconvinced that this metric accurately captures the essence of "regional memory," particularly given the limited resolution of the parcellation (only 94 nodes). Essentially, it appears akin to training a linear regression model with numerous outputs based on a single representation. This raises skepticism about the metric's ability to truly reflect regional memory dynamics. A pertinent question arises regarding how these regional memory capacities compare with node contributions in memory capacity, as computed through lesioning methods.

- The structural foundation of the reservoir layer relies on Diffusion-Weighted Imaging (DWI), yielding only positive values in the adjacency matrix. However, one of the major findings in the study suggests a correlation between the aging-related preservation of memory capacity in brain regions and the signal transduction in inhibitory neurons. Given that the RC model (with tanh activation function) lacks inhibitory links and physiological parameters representing inhibitory neurons, drawing such a conclusion becomes challenging. The assertion in the paper is primarily derived from a distinct study utilizing XGBoost, a complex machine-learning model with seemingly modest predictive performance, as evident in Fig. 2B.

Furthermore, the rationale behind this sub-project raises questions, particularly regarding the disjointed nature of the input and output set. The model is trained on a genetic dataset (i.e. Allen Human Brain Atlas (AHBA) obtained from a few subjects, while the output comprises memory capacity measures computed for a different set of subjects (to quote, "differences in regional memory capacity between the young (18-53 years old) and old (54-88 years old) individuals"). This introduces a disconnect, as the output data could encompass any information, and training XGBoost on such disparate sets might yield a semblance of predictive performance without clear relevance to the study's objectives.

- The rationale for establishing 53 years as the threshold for categorizing subjects as "old" is not explicitly elucidated. Upon examining the boxplots in Fig 1. F, G, and H, a question arises: could the significant difference in memory capacity (MC) between the two groups still be observed by raising this threshold, perhaps to 55 (the median age)? Performing an ablation study could provide valuable insights and clarity on the influence of age thresholds on observed group differences in MC.

- In Fig. 1E, despite the presence of a statistically significant group difference in the high-density regime, the magnitude of this difference (approximately 0.08) appears relatively modest. This raises questions about the hypothesis that Memory Capacity, be it in the form of a global value or a vector encompassing global and regional MCs, can serve as a robust predictor of age. Perhaps, this modest difference is reflected in Fig. 1J and K, where a rather intricate model (MLP with 4 layers, including 256, 512, 512, and 256 nodes, respectively) is necessary to predict true ages from an input vector containing 95 values (94 regional and 1 global MCs). The significance of this small value is ambiguous, and it could stem from various factors such as imaging artifacts, density thresholding, or individual variabilities. Clarifying the origin and implications of this marginal difference would strengthen the interpretation of the study's findings.

In summary, to maintain a focused and coherent narrative, I recommend streamlining the manuscript to encompass only two sub-projects: i) the exploration of memory capacity deterioration with age (Fig. 1), and ii) the investigation of the associations between global and regional memory capacities and measures of structural integrity, or other structural features. Furthermore, I suggest enhancing the depth of these investigations through the inclusion of proper control (either model-based or data-driven) studies.

(Remarks on code availability)

Reviewer #2

(Remarks to the Author)

The manuscript describes a highly interesting piece of research, applying ideas from reservoir computing and connectomics to study healthy aging, linking computational impairments with aging and altered structural and functional neuroimaging in individual participants. I have a few suggestions below, most importantly regarding the modelling process and some aspects of the analysis that I think should be clarified.

Suggestions:

Apologies if I missed it, but I did not see any optimisation of the coupling parameter in the simulations. Is it correct that a single coupling parameter was used across all individuals and simulations? If so, why is this, given that I believe it assumes a value of 1, and optimal performance in Suarez et al was achieved in a subcritical zone (coupling parameter <1). It could be the case that memory performance across the cohort could be different if coupling was adapted to optimise memory performance for each individual. This would provide interesting, more comprehensive account of the computational properties of the tractography networks.

A somewhat similar point, do we know why computational performance is different across ages? In the Suarez paper they compare models with nulls designed to keep/remove different topological properties, allowing them to attribute the altered performance to specific aspects of the network. The results from the DTI analyses suggest that reduced anisotropy within tracts might be part of the explanation; however, this could be the consequence of several factors: e.g., altered topology, selective or local reduced tract integrity or just age-related problems with image acquisition leading to more randomness. Some analysis of topological reasons behind the impaired performance would be useful.

I was quite confused by the transcriptomic analysis. It wasn't clear how the data/cross-validation etc was performed. I read the section a number of times, and was not sure what was being predicted and what left out (was it regions/genes, how was the model trained). This made it somewhat hard to work out whether statistical issues such as those related to spatial autocorrelation might affect predictive performance.

The title could be more reflective of the paper. At present, the title suggests that it is a direct measurement of memory computational capacity in empirical data, rather than derived from simulations.

A very minor point, when you first introduce the history of reservoir computing related to macroscopic neuroscience/neuroimaging in the introduction, you may also wish to include our work on this from 2017 (Hellyer et al, 2017, Plos Comp Biol).

Robert Leech

(Remarks on code availability)

Code was available, but it could have been easier to find the code relevant to the paper. I had to find the fork of the BRAPH-2 GitHub to see it (I think it was this):

<https://github.com/egolol/BRAPH-2-MemoryCapacity/tree/develop/braph2genesis/pipelines/MemoryCapacity>

Reviewer #3

(Remarks to the Author)

Mijalkov et al bring tools for AI, specifically reservoir computing, to unveil the “memory capacity” of white matter networks in a large cohort of healthy adults across a broad age range. They find interesting associations between this AI-derived read-out of the memory capacity of each individual’s connectome and their age. Several other interesting associations are also observed, including individual performance on measures of cognitive function, including memory and visual short term memory.

The topic and approach of the paper is very timely and the analyses are generally thorough and well described.

Major:

1. The individual connectomes are constructed from combining deterministic tractography by FA. The authors’ report widespread positive correlations between fractional anisotropy and global memory capacity, which is then hardly surprising (and arguably a bit circular although I guess you could argue it’s a sanity check). I would ask the authors to turn the analysis around and ask if the simpler core measure of FA (evaluated globally or in tracks via TBSS) can perform as well as the reservoir-computer measure derived from it. That would be consistent with a recent reminder that higher order measures of networks integrity should always be benchmarked against simpler one [1]. If that is the case (which seems possible), the authors could still argue that their measure of “memory” provided a link between changes in FA with age and the various “read-outs” (such as cognition etc).

2. The authors use deterministic streamline tractography applied to multi-shell diffusion-weighted data and extracting the mean fractional anisotropy (FA). It’s not clear (and I apologise if I missed it) whether the authors use FA simply to drive streamlines (as in a tensor framework) and use streamline count directly, or then go each connection and derive a measure based on the average FA along each tract? Ideally the authors would replicate their main finding by using a probabilistic approach as implemented in MrTrix 9 or any similar toolbox) because deterministic approaches have a lot of false negative connections.

3. Related to this point, are the connections in the reservoir algorithm (the matrix W on page 14) weighted or binary? If weighted, then I guess the number of edges will be the same per each density threshold but the weights will generally become weaker with age (this speaks to my main concern, point 1, that FA might alone might be a simpler predictor).

Minor:

1. “Since the initial part of the curve ($\tau = 1$ to $\tau = 5$) did not change with aging, we measured the memory capacity between delays $\tau = 6$ to $\tau = 35$ (dark-gray shaded area).” – this sounds circular – i.e. the choice of time window was made after already looking at the predicted outcome.

2. Why the same time series to all nodes? And why then randomly scaled by W (I assume this is what happens); why not just different random time series to each input node? Is this standard. I’m just lacking the background here.

3. “where tanh units” -> “were”

1. Rubinov, M. (2023). Circular and unified analysis in network neuroscience. *Elife*, 12, e79559.

(Remarks on code availability)

The code and data availability statements are appropriate. I checked the links to the code including the github repository which appear sound

Version 1:

Reviewer comments:

Reviewer #1

(Remarks to the Author)

Dear Editor,

I would like to thank the authors for conducting the complementary experiments and addressing the questions and points I raised earlier. The authors’ detailed and insightful responses and adjustments to the structure and narrative of the manuscript have clarified most of my concerns.

However, there is still one major issue that needs to be addressed before considering this work for publication. Specifically,

the method for linking genetic data and the observed age-related differences in memory capacity requires further clarification and justification.

1. Motivation:

- While the motivation has been partially explained in the response to my previous comments, it would be beneficial if the authors could elaborate on the relevance of the method in the context of their work as well as in the broader context of memory decline during the aging process. This would provide a clearer understanding of the importance and potential impact of this study.

2. Method Justification and Explanation:

- I acknowledge that regional memory capacities (MCs) reflect connectomic features and their variation during aging. However, linking genetic data with these "derived" features (i.e., differences in regional MCs that are products of another machine learning model) seems overly complex and potentially unreliable without further independent assessment and justification. Linking genetic data with "direct" features extracted from connectomic and functional data (as in references listed by authors in their response to reviewers) is typically easier to justify and interpret.

- The authors cite several studies that use methods to characterize associations between gene transcription profiles and direct features extracted from connectomes and functional networks. These studies typically use multivariate methods, such as partial least squares (PLS), to identify associations between sets of response variables and predictor variables. The method in the current manuscript differs from these established methods. The manuscript uses a different approach introduced in [1], where XGboost is employed to classify hubs. Here, the XGBoost model is employed as a regressor, which is not sufficiently justified. I understand that the size of the predictor matrix (i.e., 1158 cortical samples x 10027 genes) might have motivated using advanced machine learning models, however, the spatial resolution of the response set is much lower (i.e., only 94 nodes compared to 1158 cortical samples) and the applied up-scaling/super-resolution technique would play a critical role, specifically for the regression (compared to a classification problem in [1]).

- In the manuscript, the dimension of the predictor matrix (1158 cortical samples x 10027 genes) is only mentioned in Figure 2, while the response vector's dimension remains ambiguous. Based on Figure 2, it seems that the response vector contains differences in memory capacity, suggesting significant upscaling (~12 times larger). The explanation in Supplementary Information IX indicates that memory capacity (MC) values are assigned to cortical samples based on spatial coordinates using the AAL atlas. This could be done through value copying, linear/nonlinear extrapolations, or advanced super-resolution methods. It should be clarified.

- The Allen Human Brain Atlas (AHBA) uses a different anatomical reference, not the AAL atlas. This discrepancy introduces uncontrolled parameters to the model and interpretations.

- Despite using a sophisticated model like the XGBoost regressor, the prediction accuracy remains modest. Given the current parcellation, the XGBoost regressor, the modest prediction, and the use of two different anatomical atlases, the reliability of the method and these findings remain questionable.

Conclusion and Recommendations

While I remain critical of the method, its findings, and interpretations, if the authors wish to retain this part of the manuscript, I suggest the following:

- Summarize the method in a simpler and clearer manner to enhance reader comprehension without needing to consult additional figures and references.
- Provide a stronger justification for the use of "derived" features and the chosen machine learning approach in the context of this study.

Thank you again for your efforts. I look forward to the final version of your paper.

Ref:

[1] Xu, Zhilei, et al. "Meta-connectomic analysis maps consistent, reproducible, and transcriptionally relevant functional connectome hubs in the human brain." *Communications Biology* 5.1 (2022): 1056.

(Remarks on code availability)

The code only contains a pipeline to compute memory capacity in the reservoir computing framework. The source codes for linking genetic data and the observed age-related differences in memory capacity are missing.

Reviewer #3

(Remarks to the Author)

The authors have responded very thoroughly and with impressive additional analyses:

Studying the partial influence of age/FA as a mediator
Replicating with probabilistic tractography
Clarifying methodological steps and parameter choices

This is an interesting and important contribution to the field.

Michael Breakspear

(Remarks on code availability)

Version 2:

Reviewer comments:

Reviewer #1

(Remarks to the Author)

Dear Editor,

I appreciate the authors' efforts in making the necessary adjustments to ensure the manuscript maintains its focus on the central idea, as well as the primary analyses and findings. The revisions have greatly enhanced the clarity and impact of the work.

I have no further comments at this stage. The manuscript is well-structured and effectively presents the authors' excellent research.

Best regards,

(Remarks on code availability)

Reviewer #1 (Remarks to the Author):

In their manuscript, the authors utilize structured/spatially-embedded reservoir computing (RC) models to explore changes in the computational capacity of the brain's structural network due to aging. By comparing global and regional memory capacity (MC) between "young" and "old" subjects, the authors propose the MC metric, likely a vector of MC values, as a predictor of age. Additionally, they employ advanced machine learning techniques to establish a link between regional MCs and underlying transcriptomic profiles, along with their association with resting state functional activities and scores in five cognitive tasks.

The investigation into the contribution of connectomic features to the memory capacity of a reservoir computing model, as presented in Fig. 1, is intriguing. The results in Fig. 1F-H suggest that increasing connection density leads to a slight increase and subsequent decrease in memory capacity. Speculatively, the low-density regime results in uncontrolled connectedness, forming several small clusters with similar representations within each cluster but distinct representations between clusters. In contrast, the high-density regime, with a more connected network, exhibits less diverse representations, resulting in lower memory capacity. The medium-density range appears to strike a balance by fostering a few weakly connected clusters that guarantee sufficiently diverse representations.

Exploration of brain anatomical changes throughout aging and their computational implications are interesting. However, my enthusiasm for the work wanes due to the incorporation of multiple sub-projects. These projects divert attention from the core themes (structure, memory capacity, and aging) and instead emphasize extracting numerous features from structural, functional, and cognitive datasets. The focus shifts towards establishing correlations between these features and memory capacities. The subsequent listing of concerns is intended to address these deviations and restore clarity to the primary objectives of the study.

We thank the Reviewer for these comments and for highlighting the implications of memory capacity for aging. We understand the points raised by the Reviewer about keeping the focus of the article on its central message and we have addressed them in our revised manuscript. Below, we provide our responses to the individual comments.

- The computation of regional memory capacity, a focal point in this study, raises concerns due to its inherent ambiguity. The results section defines it as "the ability to reproduce the delayed input signal using only the activation time series of single brain regions," while the introduction states it as extending "either across all reservoir nodes (global memory capacity) or across a select subset (regional memory capacity)." A closer look at the shared codes suggests that regional memory capacity evaluates how effectively individual nodes can preserve the memory of the input. Despite comprehending the rationale behind this approach, I remain unconvinced that this metric accurately captures the essence of "regional memory," particularly given the limited resolution of the parcellation (only 94 nodes). Essentially, it appears akin to training a linear regression model with numerous outputs based on a single representation. This raises skepticism about the metric's ability to truly reflect regional memory dynamics. A pertinent question arises regarding how these regional memory capacities compare with node contributions in memory capacity, as computed through lesioning methods.

Response to comment: We have now conducted additional analyses to address the Reviewer's concerns regarding the validity of the regional memory capacity. Following the Reviewer's

suggestion, we recalculated the regional memory capacity through a lesioning approach. This lesioning method evaluates the memory capacity of a specific brain region by measuring the difference in the global memory capacity before and after removing that region as well as all of its connections from the reservoir network. We assessed the “lesional memory capacity” by computing it across the high-density range (from 21% to 30%) and calculated the area under the curve (AUC). These AUC values were used to compare the lesional memory capacity to the regional memory capacity presented in the original manuscript. To ensure consistency, we used the same input signal and input weights for the regional memory and lesional memory capacity calculations of all brain regions.

To account for the effects of age, we evaluated the association between the lesional and regional memory capacity in a reference group of 125 young participants aged under 36 years¹ from the Cam-CAN cohort. The analysis showed that the average regional values of all regions calculated by the two methods are strongly correlated with each other ($r = 0.84$, Figure R1.1A). We also calculated this correlation for all individual subjects and found that the two methods again correlated well with each other ($r = 0.73 - 0.87$, with a median value of 0.83, Figure R1.1B). These findings demonstrate that the method for calculating regional memory capacity as described in the main manuscript correlates well with the regional memory capacity derived from lesion studies, showing support for its validity to measure regional memory.

[1] Van Essen D. C., *et al.* The WU-Minn Human Connectome Project: An overview. *NeuroImage* **80**, 62–79 (2013).

Figure R1.1: Relationship between regional memory capacity and memory capacity derived from the lesioning method. A) Lesional memory capacity plotted as a function of the regional memory capacity for all regions of the AAL atlas. Both memory capacity values were calculated as an average across a reference group of 125 young individuals (< 36 years) from the Cam-CAN cohort. The orange line represents the best line of fit between the two variables. B) Histogram of the correlations between lesional and regional memory capacity for all 125 subjects in the reference group.

We revised the main manuscript and the Supplementary Appendix to add the comparison between the regional and lesional memory capacities:

- In the Manuscript, section **Results: Evaluation of the brain memory capacity** (page 4), we added the following text:

“The validity of this method to measure regional memory capacity was confirmed by an additional analysis we performed (lesional memory capacity; Supplementary Appendix: Section I), which quantifies the memory capacity of a region as the change in the network’s global memory capacity after the removal of that region and all its connections from the network.”

- In the Supplementary Appendix, we added a new section, **Section I: Comparison between regional memory capacity and memory capacity derived from lesioning studies** (page 2).
- Figure R1.1 and the associated caption was included as Figure S1 as part of this section.

- The structural foundation of the reservoir layer relies on Diffusion-Weighted Imaging (DWI), yielding only positive values in the adjacency matrix. However, one of the major findings in the study suggests a correlation between the aging-related preservation of memory capacity in brain regions and the signal transduction in inhibitory neurons. Given that the RC model (with tanh activation function) lacks inhibitory links and physiological parameters representing inhibitory neurons, drawing such a conclusion becomes challenging. The assertion in the paper is primarily derived from a distinct study utilizing XGBoost, a complex machine-learning model with seemingly modest predictive performance, as evident in Fig. 2B.

Furthermore, the rationale behind this sub-project raises questions, particularly regarding the disjointed nature of the input and output set. The model is trained on a genetic dataset (i.e., Allen Human Brain Atlas (AHBA) obtained from a few subjects, while the output comprises memory capacity measures computed for a different set of subjects (to quote, “differences in regional memory capacity between the young (18-53 years old) and old (54-88 years old) individuals”). This introduces a disconnect, as the output data could encompass any information, and training XGBoost on such disparate sets might yield a semblance of predictive performance without clear relevance to the study's objectives.

Response to comment: We thank the Reviewer for this comment. We understand the Reviewer’s concerns, which we address below by explaining better the several choices we made in this analysis.

The method we applied to link genetic data with variations in brain structure and function is a well-established approach¹ that has been used with success in several previous studies¹⁻⁴. The method employs brain-wide atlases of gene expression, which provide a spatial summary of gene transcription activity across different anatomical regions of the brain. Previous studies applied this method to establish associations between the spatial variations in gene expression across these brain-wide atlases and the spatial distribution of specific structural and functional properties of the brain evaluated in an independent sample.

Regarding the first point raised by the Reviewer, this approach does not allow to make causal inferences. It only demonstrates that the age-related differences in the regional memory capacity and the expression pattern of genes involved in communication between inhibitory neurons share a similar spatial distribution across the brain. This suggests that the preservation of memory capacity in certain regions during aging is associated with a lack of inhibition in those regions, despite memory capacity being derived only from networks with positive

connections. This interpretation is supported by previous studies on aging and brain function, which found that older individuals often over-recruit functional networks, indicating a lack of inhibition, because they need to activate more brain networks and use more neural resources than younger individuals when performing tasks^{5,6,7}. Furthermore, it is also supported by earlier findings showing that fiber tract connectivity profiles of different regions is partly governed by the profiles of gene-expression in those regions^{8,9}.

Regarding the Reviewer's second point, we would like to point out that despite the relatively modest performance of the XGBoost algorithm when a threshold of 53 years was used to define young (≤ 53 years old) and old (>53 years old) individuals, it provided replicable and robust predictions for alternative thresholds used to divide the participants. For example, when we repeated the analysis by dividing the cohort into more extreme age groups (young < 46 and old > 60 years, and young < 36 and old > 70 years), we found that the average performance of the model increased (Figure R1. 2). Additionally, the contributions of all genes to these group differences were highly correlated across the three analyses (Table R1.1). These findings demonstrate the robustness and consistency of the predictions generated by the XGBoost algorithm.

Figure R1.2: Prediction accuracy of the XGBoost algorithm for different groups.

Correlation between true and predicted age-related differences using XGBoost models and transcriptomic data. Each point represents one of the 1000 predictions in the cross-validation procedure. Age-related differences were calculated by dividing young and old individuals as follows: gray box (young ≤ 53 years, old > 53 years), orange box (young < 46 years, old > 60 years) and red box (young < 36 years, old > 70 years).

	Young <= 53 years Old > 53 years	Young < 46 years Old > 60 years	Young < 36 years Old > 70 years
Young <= 53 years Old > 53 years	1.00	0.98	0.95
Young < 46 years Old > 60 years	0.98	1.00	0.96
Young < 36 years Old > 70 years	0.95	0.96	1.000

Table R1.1: Correlation between contributions of the different genes. The correlation was evaluated by calculating the Pearson's correlation coefficient between the contributions of all genes to the prediction of age-related differences (calculated as old minus young) in the regional memory capacity.

Regarding the third point, the main rationale for using two different samples to establish a link between transcriptomics and memory capacity is based on the assumption that spatial variations in transcriptomal activity across the brain are much larger than the inter-individual variations¹. We agree with the Reviewer's concern that the validity of this assumption is not yet well understood. However, it is important to note that, since expression assays are invasive, gene expression in the brain must be measured postmortem, making it challenging to assess the inter-individual variability in these assays. There are some preliminary results that support this assumption, as noted in previous studies^{1,10}. Furthermore, this methodology has received validation by multiple studies that used it to link gene expression with structural and functional changes arising in several brain disorders. For example, it was employed to demonstrate association between previously known genes of six neurodevelopmental disorders, including Down syndrome and Velocardiofacial syndrome, to cortical anatomy changes in youth¹¹, validating its applicability. Additionally, this methodology was able to correlate regional expression of the MAPT gene with functional connectivity in Parkinson's disease¹², identify spatial expression profiles of various genes associated with white-matter dysconnectivity in schizophrenia¹³ and Huntington's disease¹⁴, understand the impact of amyloid and tau in Alzheimer's disease¹⁵, assess cortical thickness changes in children with autism spectrum disorder¹⁶, and analyze regional brain changes during neurodevelopment^{17,18}.

Finally, we used the Allen Human Brain Atlas (AHBA) to derive our results due to its extensive spatial coverage of the brain, rigorous quality control procedures, and the fact that it has been preprocessed with a unified pipeline proposed recently¹⁹.

[1] Fornito A., *et al.* Bridging the gap between connectome and transcriptome. *Trends. Cogn. Sci.* **23**(1), 34-50 (2019).

[2] Richiardi J., *et al.* Correlated gene expression supports synchronous activity in brain networks. *Science* **348**(6240), 1241-1244 (2015).

[3] Anderson K. M., *et al.* Gene expression links functional networks across cortex and striatum. *Nat Commun*, **9**(1), 1428 (2018).

[4] Vértes P. E., *et al.* Gene transcription profiles associated with inter-modular hubs and connection distance in human functional magnetic resonance imaging networks. *Philos Trans R Soc Lond B Biol Sci*, **371**(1705), 20150362 (2016).

- [5] Goldstone A., *et al.* Gender specific re-organization of resting-state networks in older age. *Front. aging neurosci.* **8**, 285 (2016).
- [6] Mijalkov M., *et al.* Sex differences in multilayer functional network topology over the course of aging in 37543 UK Biobank participants. *Netw. neurosci.* **7**(1), 351-376 (2023).
- [7] Damoiseaux J. S. Effects of aging on functional and structural brain connectivity. *Neuroimage* **160**, 32-40 (2017).
- [8] Goel P., *et al.* Spatial patterns of genome-wide expression profiles reflect anatomic and fiber connectivity architecture of healthy human brain. *Hum. Brain Mapp.* **35**(8), 4204-4218 (2014).
- [9] Forest M., *et al.* Gene networks show associations with seed region connectivity. *Hum. Brain Mapp.* **38**(6), 3126-3140 (2017).
- [10] Hawrylycz M. J., *et al.* An anatomically comprehensive atlas of the adult human brain transcriptome. *Nature* **489**(7416), 391-399 (2012).
- [11] Seidlitz J., *et al.* Transcriptomic and cellular decoding of regional brain vulnerability to neurogenetic disorders. *Nat. Commun.* **11**(1), 3358 (2020).
- [12] Rittman T., *et al.* (2016). Regional expression of the MAPT gene is associated with loss of hubs in brain networks and cognitive impairment in Parkinson disease and progressive supranuclear palsy. *Neurobiol. Aging* **48**, 153-160.
- [13] Romme I. A., *et al.* Connectome disconnectivity and cortical gene expression in patients with schizophrenia. *Biol. Psychiatry* **81**(6), 495-502 (2017).
- [14] McColgan P., *et al.* (2018). Brain regions showing white matter loss in Huntington's disease are enriched for synaptic and metabolic genes. *Biol. Psychiatry* **83**(5), 456-465.
- [15] Grothe M. J., *et al.* Molecular properties underlying regional vulnerability to Alzheimer's disease pathology. *Brain* **141**(9), 2755-2771 (2018).
- [16] Romero-Garcia R., *et al.* Synaptic and transcriptionally downregulated genes are associated with cortical thickness differences in autism. *Mol. Psychiatry* **24**(7), 1053-1064 (2019).
- [17] Kirsch L., *et al.* On expression patterns and developmental origin of human brain regions. *PLoS Comput. Biol.* **12**(8), e1005064 (2016).
- [18] Whitaker K. J., *et al.* Adolescence is associated with genomically patterned consolidation of the hubs of the human brain connectome. *Proc. Natl. Acad. Sci. U S A* **113**(32), 9105-9110 (2016).
- [19] Arnatkevičiūtė A., *et al.* A practical guide to linking brain-wide gene expression and neuroimaging data. *Neuroimage* **189**, 353-367 (2019).

We revised the manuscript in order to address the Reviewer's concerns. The changes are as follows:

- We changed the title of the relevant section in **Results** (pages 6-7) from **The spatial distribution of regional memory capacity corresponds to transcriptomics profiles**

to **Regions showing age-related changes in memory capacity overlap with regions showing hyper-excitability in aging**. This was done in order to emphasize the potential implications of this analysis as a way to interpret the observed age-related differences, instead of the stronger claim that it provides genetic basis of the memory capacity.

- We simplified the text in the relevant section (pages 6-7). The revised section focuses on the interpretation and provides support to the validity of the analysis, while not providing extensive methodological details. The revised section is as follows:

“To provide a biological interpretation of the observed age-related differences, we used a previously proposed approach²⁸ to link genetic data with neuroimaging phenotypes. This method uses brain-wide atlases of gene expression, which provide a spatial summary of gene transcription activity across the brain, and evaluates how well they overlap with the spatial distribution of the neuroimaging measure of interest. Here, we used the transcriptomic data of 10,027 genes obtained from 1,180 cortical samples from the Allen Human Brain Atlas (AHBA)^{29,30}. We then employed a XGBoost-based machine learning regression model^{31,32} to predict the spatial distribution of the differences in regional memory capacity between the young (18-53 years) and old (54-88 years) individuals (Fig. 2A). The model’s performance was evaluated by calculating the Pearson’s correlation coefficient between the true and predicted pattern of differences in regional memory capacity. Our results show that our model was able to accurately predict the pattern of age-related differences, achieving a correlation of 0.53 ± 0.05 across 1000 repetitions of the training and testing procedure (Fig. 2B).

Given the large number of genes included in the AHBA atlas, we performed a Gene Ontology (GO) enrichment analysis using GOrilla³³ to interpret these results at the level of functionally related sets of genes. After ranking all genes based on their contribution to the optimal model (Fig. 2C), this enrichment analysis aimed to identify the functional processes and cell types in which this contribution-ranked gene list was predominately expressed. Our findings, as shown in Fig. 2D and Fig. 2E, link the ability of regions to preserve memory capacity during aging to sets of functional processes involved in the inter-neuronal communication among inhibitory neurons. This suggests that dysfunctions in the communication among inhibitory neurons are associated with smaller age-related differences in memory capacity. This hyper-excitation in specific brain areas in order to diminish age-related effects could be interpreted in the context of the neuronal dedifferentiation commonly observed in older individuals, who need to activate more brain networks when compared to younger individuals to perform similar cognitive tasks^{34,35}. More details regarding the transcriptomic analysis are presented in Supplementary Appendix: Section IX.”

- We added the Reviewer’s concerns as a limitation in the manuscript (pages 12-13) as follows:

“Additionally, we used transcriptomic data obtained from only six postmortem brains to interpret our results regarding the age-related differences across the lifespan. Such interpretations are based on the assumption that spatial variations in transcriptomic activity across the brain are much larger than the inter-individual variations²⁸, however, the validity of this assumption is not yet completely understood. It is also important to note that this analysis does not establish causality between genetic factors and memory capacity, but rather provides information regarding the spatial overlap between the two. However, despite these limitations, this method has been successfully used in previous studies to associate known

genetic factors with the spatial distribution of neuroimaging markers in several disorders^{75–77}, which demonstrates that the transcriptomic analysis can provide valuable and novel insights for interpreting our results.”

- The following references were added to the manuscript:

28. Fornito, A., Arnatkevičiūtė, A. & Fulcher, B. D. Bridging the Gap between Connectome and Transcriptome. *Trends Cogn. Sci.* **23**, 34–50 (2019).

75. Seidlitz, J. *et al.* Transcriptomic and cellular decoding of regional brain vulnerability to neurogenetic disorders. *Nat. Commun.* **11**, 3358 (2020).

76. Romme, I. A. C., de Reus, M. A., Ophoff, R. A., Kahn, R. S. & van den Heuvel, M. P. Connectome Disconnectivity and Cortical Gene Expression in Patients With Schizophrenia. *Biol. Psychiatry* **81**, 495–502 (2017).

77. Romero-Garcia, R., Warrier, V., Bullmore, E. T., Baron-Cohen, S. & Bethlehem, R. A. I. Synaptic and transcriptionally downregulated genes are associated with cortical thickness differences in autism. *Mol. Psychiatry* **24**, 1053–1064 (2019).

- In the Supplementary Appendix, we added a new section, **Section IX: Methodological considerations of transcriptomics analysis** (pages 9-22) which provides more detailed account of the methodology and results of this analysis.
- Figure R1.2 and the associated caption was included as Figure S8 as part of this section. Table R1.1 was also included as Table S5.

- The rationale for establishing 53 years as the threshold for categorizing subjects as "old" is not explicitly elucidated. Upon examining the boxplots in Fig 1. F, G, and H, a question arises: could the significant difference in memory capacity (MC) between the two groups still be observed by raising this threshold, perhaps to 55 (the median age)? Performing an ablation study could provide valuable insights and clarity on the influence of age thresholds on observed group differences in MC.

Response to comment: We chose 53 years as an age threshold to divide individuals into young and old because 53 years is the mean age of the Cam-Can lifespan cohort, which includes individuals from 18 to 88 years old. Following the Reviewer's suggestion, we repeated our analysis by dividing the cohort into young and old individuals using age thresholds ranging from 39 to 70 years and compared the memory capacity between groups. This range was chosen in order to ensure that the smaller group comprised at least 25% of the subjects in the cohort in each case. Our results demonstrated that the old individuals had lower memory capacity compared to young ones across the complete range (Figure R1.3A and Figure R1.3B), which was significant across the entire range of thresholds ($p_{\max} < 10^{-4}$).

Figure R1.3: Memory capacity as a function of the age threshold used to define young and old individuals. **A)** Average memory capacity for young (orange circles) and old (red circles) individuals as a function of the age-threshold used to categorize subjects into these groups. The error bars represent one standard error of the mean for each group. **B)** Between-group differences (calculated as old minus young) as a function of the age-threshold.

We added these results as follows:

- In the Manuscript, section **Results: The connectome of older individuals shows lower computational memory capacity** (page 5), we added the following text:

“Furthermore, they remained unchanged after changing the threshold of 53 years by several different thresholds (39 – 70 years) to place individuals in the young and old groups (Supplementary Appendix: Section IV).”

- In the Supplementary Appendix, we added a new section, **Section IV: Differences in memory capacity as a function of age threshold** (pages 4-5).
- Figure R1.3 and the associated caption was included as Figure S4 as part of this section.

- In Fig. 1E, despite the presence of a statistically significant group difference in the high-density regime, the magnitude of this difference (approximately 0.08) appears relatively modest. This raises questions about the hypothesis that Memory Capacity, be it in the form of a global value or a vector encompassing global and regional MCs, can serve as a robust predictor of age. Perhaps, this modest difference is reflected in Fig. 1J and K, where a rather intricate model (MLP with 4 layers, including 256, 512, 512, and 256 nodes, respectively) is necessary to predict true ages from an input vector containing 95 values (94 regional and 1 global MCs). The significance of this small value is ambiguous, and it could stem from various factors such as imaging artifacts, density thresholding, or individual variabilities. Clarifying the origin and implications of this marginal difference would strengthen the interpretation of the study's findings.

Response to comment: The seemingly small differences in memory capacity values arise from our use of 53 years as the threshold to categorize the individuals into young and old groups. Consequently, the two groups also include individuals with ages that are close to this threshold,

contributing to the perceived small differences. To address this, we conducted additional analyses by comparing more extreme age groups. Starting from a threshold of 53 years (as presented in the main manuscript), we progressively increased the threshold by 1 year for older subjects and decreased it by 1 year for younger ones, ensuring symmetry around the age of 53 years. The most extreme groups that we analyzed included individuals younger than 36 and older than 70. We then compared the means of memory capacity at high-density ranges between these two extreme groups and calculated Cohen's *d* to measure the effect size of these differences.

Figure R1.4: Memory capacity in more extreme groups. Differences in memory capacity between old and young individuals as a function of network density for different thresholds to define old (young) groups **A)** 55 (51), **B)** 60 (46), and **C)** 70 (36). **D)** Differences in high-density memory capacity (calculated as old minus young) as a function of the age gap between old and young individuals. The age gap was centered around the age of 53 years. **E)** Cohen's *d* value as a function of the age-gap between the young and old individuals.

Figure R1.4 A-C shows the between-group differences for three thresholds as a function of density. The differences followed the same pattern as Fig. 1E in the main manuscript. However, while the difference magnitude was 0.08 for less extreme groups (Figure R1.4 A), it increased to 0.11 as the threshold to define the groups was increased (Figure R1.4 B-C). This increasing

magnitude of difference between old and young individuals was more evident when evaluating the AUC of the high-density range (Figure R1.4 D), which remained statistically significant for all age gaps ($p_{\max} < 10^{-4}$). Finally, the Cohen’s d values calculated for each comparison in Figure R1.4 D ranged between 0.81 and 0.85, with a median value of 0.83 (Figure R1.4E). These analyses show that, despite being seemingly small in magnitude, the mean differences have large effect sizes (Cohen’s $d > 0.8$)¹, supporting the hypothesis that memory capacity is a good age-related biomarker.

Furthermore, we assessed how our prediction of individual age changed when using multilayer perceptron models with different complexities. To achieve this, we modified the number of nodes in all layers and reevaluated the model’s prediction accuracy. Our findings (Table R1.2) showed that the model presented in the main manuscript, with a layer configuration of [256, 512, 512, 256], had the optimal performance. However, high correlation between true and predicted age could also be achieved with simpler model configurations, reaching an average correlation coefficient of 0.64 for a simple 1 neuron model.

Complexity of multilayer perceptron model	Correlation between true and predicted age
[1]	0.64 ± 0.07
[4, 8, 8, 4]	0.63 ± 0.12
[8, 16, 16, 8]	0.67 ± 0.06
[16, 32, 32, 16]	0.69 ± 0.05
[32, 64, 64, 32]	0.69 ± 0.05
[64, 128, 128, 64]	0.72 ± 0.04
[128, 256, 256, 128]	0.74 ± 0.04
[256, 512, 512, 256]	0.74 ± 0.05
[512, 1024, 1024, 512]	0.73 ± 0.04
[1024, 2048, 2048, 1024]	0.71 ± 0.04

Table R1.2: Performance of alternative deep-learning models with varying complexity for individual age prediction. Correlation between true and predicted age as a function of different deep-learning multilayer perceptron models (ranging from simpler to more complex from top to bottom). The average and the standard deviation values were derived by 10-fold cross-validation procedure.

[1] Cohen, J. (2013). *Statistical power analysis for the behavioral sciences*. Routledge.

We added these analyses in the manuscript as follows:

- In the Manuscript, section **Results: The connectome of older individuals shows lower computational memory capacity** (page 5), we added the following text:

“The differences had a large effect size (Cohen’s $d > 0.8$) and were even more pronounced when comparing more extreme age groups, i.e., with increasing age gaps between the two groups (Supplementary Appendix: Section III).”

- In the Manuscript, section **Results: Computational memory capacity is an excellent predictor of aging** (page 6), we added the following text:

“More details about model selection are shown in Supplementary Appendix: Section VIII.”

- In the Supplementary Appendix, we added two new sections, **Section III: Differences in memory capacity between young and old individuals for extreme age groups**

(pages 3-4) and **Section VIII: Age prediction using deep-learning models of varying complexity** (page 9).

- Figure R1.4 and the associated caption was included as Figure S3 as part of section III. Table R1.2 was included as Table S1 as part of section VIII.

In summary, to maintain a focused and coherent narrative, I recommend streamlining the manuscript to encompass only two sub-projects: i) the exploration of memory capacity deterioration with age (Fig. 1), and ii) the investigation of the associations between global and regional memory capacities and measures of structural integrity, or other structural features. Furthermore, I suggest enhancing the depth of these investigations through the inclusion of proper control (either model-based or data-driven) studies.

Response to comment: We thank the Reviewer for the suggestions on how to improve our manuscript. We have implemented the comments and reorganized the manuscript in order to present a more focused and coherent narrative. The changes we implemented are as follows:

1. We divided the “Results” section into two separate subsections: “Memory capacity during aging” (pages 5-7) and “Associations between memory capacity, neuroimaging measures of brain structure and function, and cognition” (pages 7-9). The first section contains the results of the study of the memory capacity deterioration with age. It comprises the subsections that describe how the global and regional memory capacity change with aging, the prediction of individual age using deep-learning multilayer perceptron models and the genetic results which provide biological interpretation of the observed regional age-related differences. The second section summarizes our findings of the associations of memory capacity with anatomical connectivity, functional activation, signal intensity of locus coeruleus and cognition.
2. We followed the Reviewer’s suggestion and simplified the presentation of genetic analysis results throughout the manuscript. The relevant section has been condensed and the full details of the analysis are now presented in the Supplementary Appendix, Section IX. We did not want to eliminate this section completely because it can still provide valuable insights and possible interpretation of the age-related differences we observe. However, we have revised the manuscript to highlight the limitations of this analysis.
3. We also increased the depth of our investigations of memory capacity by performing several additional studies aimed to demonstrate how memory capacity is associated with other measures of brain structure and network topology that serve as control measures. Altogether, our results demonstrated that the memory capacity contained information beyond the information provided by simpler measures of brain structure and network topology, showing that memory capacity offers unique insights into aging effects beyond the ones provided by simpler measures. Specifically, we conducted these additional studies:
 - 3.1. We compared memory capacity with the simpler measure of mean global fractional anisotropy (FA), as well as the individual FA values of several anatomical tracts. Our findings showed that memory capacity can explain variance in age beyond the variance that can be attributed to the FA values.
 - 3.2. We examined the association of memory capacity with measures of network integration (global efficiency) and segregation (clustering coefficient), as well as with the average weight of the network after normalization. Our results showed

that the decrease of memory capacity with age was significantly associated with lower global efficiency and lower average network weight, but not with clustering coefficient. However, when conducting mediation analyses to assess how these measures affect the relationship between memory capacity and age, we found that global efficiency and clustering coefficient were not significant mediators, while network weight was only a weak mediator, mediating only 8.8% of the relation between memory capacity and age.

- 3.3. To assess the reliability of the memory capacity, we showed that the measure is robust against changing the normalization parameters, as well as against different preprocessing pipelines.

We revised the manuscript to include these additional analyses:

- In section **Results: The connectome of older individuals shows lower computational memory capacity** (pages 5-6), we added the following text:

“Moreover, we conducted additional analyses to assess whether topological measures or FA values explained the relationship between memory capacity and age. Our findings showed that, while these measures partially mediated the relationship, memory capacity provided independent information about connectome changes throughout aging (Supplementary Appendix: Section V, Section VI, and Section VII).”

- In section **Methods: Reservoir computing** (page 14), we added the following text:

“however, we found that the performance of the model was robust against the variation of this parameter. Further details regarding the choice of the normalization parameter are presented in Supplementary Appendix: Section XIII.”

- In section **Methods: Calculation of whole-brain anatomical connectivity networks** (page 18), we added the following text:

“To evaluate the robustness of our findings against different preprocessing pipelines, we replicated our results using probabilistic tractography to build the anatomical connectivity networks (Supplementary Appendix: Section XIV).”

- The details of these results were summarized by adding several new sections in the Supplementary Appendix. The sections we added are: **Section V: Relationship between memory capacity and measures of network topology and brain structural integrity** (pages 5-7), **Section VI: Mediation analysis of the relationship between memory capacity and age** (page 7), **Section VII: Relationship between age, memory capacity and FA values of individual tracts** (pages 7-8), **Section XIII: Normalization procedure for the whole-brain anatomical networks** (pages 25-26), and **Section XIV: Robustness against different preprocessing pipelines** (pages 26-27).

Reviewer #2 (Remarks to the Author):

The manuscript describes a highly interesting piece of research, applying ideas from reservoir computing and connectomics to study healthy aging, linking computational impairments with aging and altered structural and functional neuroimaging in individual participants. I have a few suggestions below, most importantly regarding the modelling process and some aspects of the analysis that I think should be clarified.

We thank the Reviewer for the positive comments on our work, and for recognizing its potential impact on understanding aging and age-associated changes in brain function and structure. The revised manuscript addresses and incorporates all of the Reviewer's suggestions. We provide our point-by-point responses to the individual comments below.

Suggestions:

Apologies if I missed it, but I did not see any optimisation of the coupling parameter in the simulations. Is it correct that a single coupling parameter was used across all individuals and simulations? If so, why is this, given that I believe it assumes a value of 1, and optimal performance in Suarez et al was achieved in a subcritical zone (coupling parameter < 1). It could be the case that memory performance across the cohort could be different if coupling was adapted to optimise memory performance for each individual. This would provide interesting, more comprehensive account of the computational properties of the tractography networks.

Response to comment: We thank the Reviewer for this comment. In our work, we normalized the weighted connectivity matrices to ensure that the largest singular value, denoted by s_{SVD} , of all matrices was equal to unity. This is a standard procedure¹ that allows the reservoir dynamics to synchronize with the input signal. Therefore, we did not use a single coupling parameter across the sample, instead, we normalized the largest singular value for each participant and density independently.

The normalization was needed in order to achieve the echo-state property, which is a necessary condition for the reproduction of time series¹. A recent paper outlined the procedure for choosing reservoir parameters to achieve this property². As demonstrated by the authors of Ref. 2, when “tanh” is used as an activation function for the reservoir nodes, normalizing the largest singular value of the reservoir's connectivity matrix to unity is a sufficient condition that guarantees the presence of the echo-state property throughout the simulation.

Following the Reviewer's suggestion, we conducted additional analyses to assess different parameters for this normalization. In these analyses, the normalization was performed by scaling the connectivity matrices (for each participant and at each density) such that their largest singular value s_{SVD} was in the range 0.1 to 1.5. Furthermore, since larger values of $s_{SVD} < 1$ result in longer memory length³, we further subdivided the range 0.9 to 1.0. Table R2.1 shows the global memory capacity for all values of the normalization parameter across the complete cohort of individuals. We observed that the highest global memory capacity can be obtained at $s_{SVD} = 0.96$. However, when evaluating the age-prediction ability with the deep-learning multilayer perceptron model, we found that both $s_{SVD} = 0.96$ and $s_{SVD} = 1$, provided similar prediction ability, with average correlation between predicted and true age across 10-fold cross-validation being 0.72 ± 0.04 and 0.72 ± 0.03 , respectively.

Maximum singular value (s_{SVD})	Global memory capacity (high-density)	Maximum singular value (s_{SVD})	Global memory capacity (high-density)
0.1	0.77 - 0.98(0.87)	0.94	70.45 - 73.84(72.22)
0.2	4.09 - 5.08(4.56)	0.95	71.13 - 74.39(72.74)
0.3	12.95 - 14.17(13.60)	0.96	71.30 - 74.35(72.86)
0.4	20.53 - 22.55(21.59)	0.97	70.76 - 74.13(72.43)
0.5	27.63 - 29.52(28.67)	0.98	69.90 - 72.87(71.30)
0.6	36.32 - 38.55(37.40)	0.99	67.82 - 70.95(69.34)
0.7	44.73 - 46.95(45.96)	1.0	63.83 - 68.37(65.74)
0.8	54.22 - 57.44(56.00)	1.1	47.42 - 51.54(49.08)
0.9	66.29 - 69.99(68.02)	1.2	39.84 - 42.61(40.78)
0.91	67.62 - 71.16(69.25)	1.3	33.92 - 36.25(34.75)
0.92	68.87 - 72.06(70.40)	1.4	29.34 - 31.27(30.05)
0.93	69.89 - 73.12(71.41)	1.5	26.28 - 27.88(26.93)

Table R2.1: Optimization of normalization parameters for the reservoir connectivity matrix. We calculated the global memory capacity for the cases when the largest singular value of each network, s_{SVD} , is normalized in the range 0.1 to 1.5. The table shows the minimum and maximum memory capacity (median in parenthesis) across all individuals in the Cam-CAN cohort.

[1] Lukoševičius, M. & Jaeger, H. Reservoir computing approaches to recurrent neural network training. *Comput. Sci. Rev.* **3**, 127–149 (2009).

[2] Storm L., *et al.* Constraints on parameter choices for successful time-series prediction with echo-state networks. *Mach. learn.: sci. technol.* **3**(4), 045021 (2022).

[3] Gallicchio C., *et al.* Deep reservoir computing: A critical experimental analysis. *Neurocomputing* **268**, 87-99 (2017).

We revised the main manuscript and the Supplementary Appendix to add these additional analyses:

- In the Manuscript, section **Methods: Reservoir computing** (page 14), we added the following text:

“however, we found that the performance of the model was robust against the variation of this parameter. Further details regarding the choice of the normalization parameter are presented in Supplementary Appendix: Section XIII.”

- In the Supplementary Appendix, we added a new section, **Section XIII: Normalization procedure for the whole-brain anatomical networks** (pages 25-26), where we explain the rationale behind the normalization procedure we used in the main analyses.
- Table R2.1 was added as Table S6 in Supplementary Section XIII.

A somewhat similar point, do we know why computational performance is different across ages? In the Suarez paper they compare models with nulls designed to keep/remove different topological properties, allowing them to attribute the altered performance to specific aspects of the network. The results from the DTI analyses suggest that reduced anisotropy within tracts might be part of the explanation; however, this could be the consequence of several factors:

e.g., altered topology, selective or local reduced tract integrity or just age-related problems with image acquisition leading to more randomness. Some analysis of topological reasons behind the impaired performance would be useful.

Response to comment: Following the Reviewer's suggestion, in order to understand why computational performance is different across ages, we assessed the relation between memory capacity, different measures of network topology, and the mean fractional anisotropy values.

To evaluate the topology of the network, we calculated the global efficiency and clustering coefficients, which are measures of network integration (the ability with which nodes communicate with each other through direct, efficiency connections) and network segregation (representing the clustered connectivity across all network nodes), respectively. Both measures were calculated on binary networks, using the BRAPH software¹. Furthermore, we also evaluated the average connectivity weight of the network, as well as the mean FA values for each individual. We note that the average weight was calculated after the normalization of the connectivity matrices to ensure their largest singular value is equal to one. Global efficiency, clustering coefficient and average weight were calculated for each density in the high-density range (21% to 30%). The corresponding area under the curve (AUC) across this range was obtained by integrating these values and used for comparison to the high-density global memory capacity.

We found that high memory capacity was associated with high global efficiency ($r = 0.11$, $p = 0.005$, Figure R2.1A), high average weight ($r = 0.37$, $p < 0.001$, Figure R2.1C) and mean FA values ($r = 0.32$, $p < 0.001$, Figure R2.1D). There was no significant association between the global memory capacity and network clustering coefficient (Figure R2.1B). This demonstrates that the changes in computational ability throughout aging are associated with the ability of brain regions to communicate directly with each other and the average strength of anatomical connectivity between them. To understand these associations further, we conducted a mediation analysis² and investigated whether these topological measures affect the relation between memory capacity and age across the lifespan. We found that clustering coefficient and global efficiency were not significant mediators of the relationship between memory capacity and age ($p = 0.56$ and $p = 0.19$). However, mean fractional anisotropy (FA) values and average weight were significant mediators, where mean FA values mediated 53.2% ($p < 0.001$, Figure R2.2A), while the average normalized weight mediated only 8.8% ($p = 0.04$) of the relationship between memory capacity and age (Figure R2.2B).

Altogether, these analyses demonstrate that the age-related differences in computational ability are partly mediated by the mean FA values and only weakly mediated by the average normalized network weight. Importantly, they highlight the fact that memory capacity provides independent information about how the connectome changes throughout aging, which is complementary to the information provided by alternative measures.

Figure R2.1: Relationship between memory capacity, measures of network topology, and global mean FA values. Associations between global memory capacity and **A)** global efficiency, **B)** clustering coefficient, **C)** normalized average network weight and **D)** mean FA values. Each circle represents an individual subject of the Cam-CAN cohort; the correlations were evaluated across the entire cohort. The orange lines represent the best line of fit between the two variables.

A) Normalized average weight as mediator B) Mean FA values as mediator

Figure R2.2: Mediation analysis. Mediation analysis of the relationship between memory capacity and age with **A)** normalized average weight and **B)** mean FA values as mediators. Plots are showing effect estimates for the indirect (mediation), direct and total effect, where percentage of mediation was calculated as Indirect effect / Total effect * 100%.

[1] Mijalkov M., *et al.* BRAPH: A graph theory software for the analysis of brain connectivity. *PLOS ONE* **12**, e0178798 (2017).

[2] Tingley D., *et al.* mediation: R Package for Causal Mediation Analysis. *J. Stat. Softw.* **59**(5), 1–38 (2014)

We modified the main manuscript to add these analyses. The changes are as follows:

- In section **Results: The connectome of older individuals shows lower computational memory capacity** (pages 5-6), we added the following text:

“Moreover, we conducted additional analyses to assess whether topological measures or FA values explained the relationship between memory capacity and age. Our findings showed that, while these measures partially mediated the relationship, memory capacity provided independent information about connectome changes throughout aging (Supplementary Appendix: Section V, Section VI, and Section VII).”

- In the Supplementary Appendix, we added two new sections, **Section V: Relationship between memory capacity and measures of network topology and brain structural integrity** (pages 5-7) and **Section VI: Mediation analysis of the relationship between memory capacity and age** (page 7), in which we detail the methodology for these analyses.
- Figure R2.1 and its associated caption was included as Figure S5 in section V. Additionally, Figure R2.2 was included as Figure S6 in section VI.

I was quite confused by the transcriptomic analysis. It wasn't clear how the data/cross-validation etc was performed. I read the section a number of times, and was not sure what was being predicted and what left out (was it regions/genes, how was the model trained). This made it somewhat hard to work out whether statistical issues such as those related to spatial autocorrelation might affect predictive performance.

Response to comment: We thank the Reviewer for this comment. We used the transcriptomics analysis to study associations between the spatial variation of gene-expression profiles across the brain and the spatial variations in given neuroimaging phenotypes of brain structure or function¹.

We used the Allen Human Brain Atlas (AHBA) to obtain transcriptomic data across thousands of genes in every region of the brain parcellation. We decided to use this atlas because of its anatomically comprehensive spatial coverage of the brain, its documented quality-control procedures, and the availability of a unified preprocessing pipeline¹⁻³. AHBA provides gene expression data for 1,285 cortical tissue samples (across 10,027 genes) across the brain.

Our analysis related this spatial distribution of gene-expression data to the spatial pattern of regional age-related differences in memory capacity. These age-related differences were calculated for 94 regions of the AAL brain atlas used in our study. Using the MNI coordinates of each brain sample provided by the AHBA gene-expression atlas, we mapped the 1,180 brain samples to the corresponding regions of the AAL atlas (and to the difference in memory capacity in that brain region), while we did not use the unmatched 105 samples in the analysis. Finally, we used the XGBoost algorithm to predict the differences in memory capacity values in each brain sample from the gene-expression values for that sample.

Therefore, in the cross-validation procedure we removed brain samples. Specifically, we randomly split the 1,180 samples into 1,000 samples for training and 180 samples for testing. The cross-validation procedure was performed in two steps. In step 1, it was used as a pre-training procedure, which used the 1,000 training sample in a 10-fold cross-validation procedure to determine the number of optimal iterations. In the following step, the number of optimal iterations, together with the 1,000 training samples, were used to obtain an optimal model, and the contribution of each gene to this model. Finally, the predictive power of the resulting optimal model was tested using the 180 testing samples. Since the number of genes included in the AHBA atlas is large, to interpret our results we focused on the level of functionally related set of genes based on Gene Ontology (GO) enrichment analysis^{1,4}, rather than focusing on individual genes.

Additionally, the Reviewer is correct in that spatial autocorrelation has been recognized as a potential issue in this analysis¹. However, despite this issue, the method has received widespread validation in many studies using it to associate previously known genes with spatial distributions of several measures of brain structure and function in several disorders⁵⁻¹⁰. Specifically, a previous study investigated the problem of spatial autocorrelation, and demonstrated that analyses using XGBoost and AHBA dataset, as we do in the current manuscript, produce results with multifaceted neural relevance, rather than being artifacts of spatial autocorrelations¹¹.

[1] Fornito A., *et al.* Bridging the gap between connectome and transcriptome. *Trends. Cogn. Sci.* **23**(1), 34-50 (2019).

[2] Hawrylycz M. J., *et al.* An anatomically comprehensive atlas of the adult human brain transcriptome. *Nature* **489**(7416), 391-399 (2012).

[3] Arnatkevičiūtė A., *et al.* A practical guide to linking brain-wide gene expression and neuroimaging data. *Neuroimage* **189**, 353-367 (2019).

[4] Ashburner M., *et al.* Gene ontology: tool for the unification of biology. *Nat. Genet.*, **25**(1), 25-29 (2000).

[5] Seidlitz J., *et al.* Transcriptomic and cellular decoding of regional brain vulnerability to neurogenetic disorders. *Nat. Commun.* **11**(1), 3358 (2020).

[6] Rittman T., *et al.* (2016). Regional expression of the MAPT gene is associated with loss of hubs in brain networks and cognitive impairment in Parkinson disease and progressive supranuclear palsy. *Neurobiol. Aging* **48**, 153-160.

[7] Romme I. A., *et al.* Connectome disconnectivity and cortical gene expression in patients with schizophrenia. *Biol. Psychiatry* **81**(6), 495-502 (2017).

[8] McColgan P., *et al.* (2018). Brain regions showing white matter loss in Huntington's disease are enriched for synaptic and metabolic genes. *Biol. Psychiatry* **83**(5), 456-465.

[9] Grothe M. J., *et al.* Molecular properties underlying regional vulnerability to Alzheimer's disease pathology. *Brain* **141**(9), 2755-2771 (2018).

[10] Romero-Garcia R., *et al.* Synaptic and transcriptionally downregulated genes are associated with cortical thickness differences in autism. *Mol. Psychiatry* **24**(7), 1053-1064 (2019).

[11] Xu Z., *et al.* Meta-connectomic analysis maps consistent, reproducible, and transcriptionally relevant functional connectome hubs in the human brain. *Commun. Biol.* **5**, 1056 (2022).

We thoroughly revised the manuscript to address the Reviewer's comment. Specifically, we applied the following changes:

- The title of the section describing the transcriptomic analysis (in **Results**, pages 6-7) has been changed from **The spatial distribution of regional memory capacity corresponds to transcriptomics profiles** to **Regions showing age-related changes in memory capacity overlap with regions showing hyper-excitability in aging**. We believe that this emphasizes the potential implications of this analysis as a way to interpret the observed age-related differences.
- We simplified the text in the relevant section (pages 6-7). In the revised section, the text focuses on the interpretation of the obtained results and offers more support to the validity of the method. The section does not provide any specific details about the methodology, which have now been moved to the Supplementary Appendix. The revised section is as follows:

“To provide a biological interpretation of the observed age-related differences, we used a previously proposed approach²⁸ to link genetic data with neuroimaging phenotypes. This method uses brain-wide atlases of gene expression, which provide a spatial summary of gene transcription activity across the brain, and evaluates how well they overlap with the spatial distribution of the neuroimaging measure of interest. Here, we used the transcriptomic data of 10,027 genes obtained from 1,180 cortical samples from the Allen Human Brain Atlas (AHBA)^{29,30}. We then employed a XGBoost-based machine learning regression model^{31,32} to predict the spatial distribution of the differences in regional memory capacity between the young (18-53 years) and old (54-88 years) individuals (Fig. 2A). The model's performance

was evaluated by calculating the Pearson's correlation coefficient between the true and predicted pattern of differences in regional memory capacity. Our results show that our model was able to accurately predict the pattern of age-related differences, achieving a correlation of 0.53 ± 0.05 across 1000 repetitions of the training and testing procedure (Fig. 2B).

Given the large number of genes included in the AHBA atlas, we performed a Gene Ontology (GO) enrichment analysis using GOrilla³³ to interpret these results at the level of functionally related sets of genes. After ranking all genes based on their contribution to the optimal model (Fig. 2C), this enrichment analysis aimed to identify the functional processes and cell types in which this contribution-ranked gene list was predominately expressed. Our findings, as shown in Fig. 2D and Fig. 2E, link the ability of regions to preserve memory capacity during aging to sets of functional processes involved in the inter-neuronal communication among inhibitory neurons. This suggests that dysfunctions in the communication among inhibitory neurons are associated with smaller age-related differences in memory capacity. This hyper-excitation in specific brain areas in order to diminish age-related effects could be interpreted in the context of the neuronal dedifferentiation commonly observed in older individuals, who need to activate more brain networks when compared to younger individuals to perform similar cognitive tasks^{34,35}. More details regarding the transcriptomic analysis are presented in Supplementary Appendix: Section IX.”

- We added a limitation section in the manuscript (page 13) that highlights possible limitations of the analysis that need to be considered:

“Additionally, we used transcriptomic data obtained from only six postmortem brains to interpret our results regarding the age-related differences across the lifespan. Such interpretations are based on the assumption that spatial variations in transcriptomic activity across the brain are much larger than the inter-individual variations²⁸, however, the validity of this assumption is not yet completely understood. It is also important to note that this analysis does not establish causality between genetic factors and memory capacity, but rather provides information regarding the spatial overlap between the two. However, despite these limitations, this method has been successfully used in previous studies to associate known genetic factors with the spatial distribution of neuroimaging markers in several disorders^{75–77}, which demonstrates that the transcriptomic analysis can provide valuable and novel insights for interpreting our results.”

- The following references were added to the manuscript:

28. Fornito, A., Arnatkevičiūtė, A. & Fulcher, B. D. Bridging the Gap between Connectome and Transcriptome. *Trends Cogn. Sci.* **23**, 34–50 (2019).

75. Seidlitz, J. *et al.* Transcriptomic and cellular decoding of regional brain vulnerability to neurogenetic disorders. *Nat. Commun.* **11**, 3358 (2020).

76. Romme, I. A. C., de Reus, M. A., Ophoff, R. A., Kahn, R. S. & van den Heuvel, M. P. Connectome Disconnectivity and Cortical Gene Expression in Patients With Schizophrenia. *Biol. Psychiatry* **81**, 495–502 (2017).

77. Romero-Garcia, R., Warrier, V., Bullmore, E. T., Baron-Cohen, S. & Bethlehem, R. A. I. Synaptic and transcriptionally downregulated genes are associated with cortical thickness differences in autism. *Mol. Psychiatry* **24**, 1053–1064 (2019).

- In the Supplementary Appendix, we added a new section, **Section IX: Methodological considerations of transcriptomics analysis** (pages 9-22). This section provides more details about the methodology and results of this analysis. Specifically, the new Section includes a paragraph describing how we account for spatial autocorrelation, which was mentioned by the Reviewer as possible limitation that might affect the predictive performance. The relevant paragraph (page 10) is as follows:

“A potential issue that could affect the predictive performance of this model is spatial autocorrelation, which states that regions that are closer to each other tend to have more similar transcriptional profiles⁷. To account for this issue, we followed the statistical approach presented in an earlier study¹³, which investigated the issue of spatial autocorrelation and showed that analyses combining XGBoost and AHBA dataset obtain findings that have neural relevance, rather than being artifacts of spatial autocorrelations.”

- The relevant references added to the Supplementary Appendix are:

[7] Fornito, A., Arnatkevičiūtė, A. & Fulcher, B. D. Bridging the Gap between Connectome and Transcriptome. *Trends Cogn. Sci.* **23**, 34–50 (2019).

[13] Xu, Z. *et al.* Meta-connectomic analysis maps consistent, reproducible, and transcriptionally relevant functional connectome hubs in the human brain. *Commun. Biol.* **5**, 1–17 (2022).

The title could be more reflective of the paper. At present, the title suggests that it is a direct measurement of memory computational capacity in empirical data, rather than derived from simulations.

Response to comment: We thank the Reviewer for this suggestion. We changed the title to: **“Computational memory capacity predicts aging and cognitive decline”**.

A very minor point, when you first introduce the history of reservoir computing related to macroscopic neuroscience/neuroimaging in the introduction, you may also wish to include our work on this from 2017 (Hellyer et al, 2017, Plos Comp Biol).

Response to comment: We apologize for missing this reference. We included the reference in the revised version of our manuscript as follows:

“Here we use reservoir computing¹⁰-- a machine-learning framework that can reproduce the functional dynamics of biological networks¹¹⁻¹⁴ -- to characterize how the cognitive function of the human brain deteriorates with aging.”

14. Hellyer, P. J., Clopath, C., Kehagia, A. A., Turkheimer, F. E. & Leech, R. From homeostasis to behavior: Balanced activity in an exploration of embodied dynamic environmental-neural interaction. *PLOS Comput. Biol.* **13**, e1005721 (2017).

Robert Leech

Reviewer #2 (Remarks on code availability):

Code was available, but it could have been easier to find the code relevant to the paper. I had to find the fork of the BRAPH-2 GitHub to see it (I think it was this):

<https://github.com/egolol/BRAPH-2-MemoryCapacity/tree/develop/braph2genesis/pipelines/MemoryCapacity>

Response to comment: Following the Reviewer's suggestion, we changed the location of the code and provided a more detailed explanation on where to find the code relevant to the paper. The code now can be accessed at: <https://github.com/braph-software/MemoryCapacity>.

Reviewer #3 (Remarks to the Author):

Mijalkov et al bring tools for AI, specifically reservoir computing, to unveil the “memory capacity” of white matter networks in a large cohort of healthy adults across a broad age range. They find interesting associations between this AI-derived read-out of the memory capacity of each individual’s connectome and their age. Several other interesting associations are also observed, including individual performance on measures of cognitive function, including memory and visual short term memory.

The topic and approach of the paper is very timely and the analyses are generally thorough and well described.

We thank the Reviewer for highlighting the strengths and potential impact of our analyses. We have addressed the Reviewer’s comments in the revised manuscript and our responses at the individual comments follow below.

Major:

1. The individual connectomes are constructed from combining deterministic tractography by FA. The authors’ report widespread positive correlations between fractional anisotropy and global memory capacity, which is then hardly surprising (and arguably a bit circular although I guess you could argue it’s a sanity check). I would ask the authors to turn the analysis around and ask if the simpler core measure of FA (evaluated globally or in tracks via TBSS) can perform as well as the reservoir-computer measure derived from it. That would be consistent with a recent reminder that higher order measures of networks integrity should always be benchmarked against simpler one [1]. If that is the case (which seems possible), the authors could still argue that their measure of “memory” provided a link between changes in FA with age and the various “read-outs” (such as cognition etc).

Response to comment: We thank the Reviewer for this suggestion. Indeed, the core measure of mean FA shows a high correlation with age in our sample. Following the suggestion, we conducted several additional analyses to demonstrate that the variation of memory capacity with age provides information that goes beyond the effect of the measure of FA. These additional analyses were performed using the mean global FA values, as well as the FA in the following left (L) and right (R) tracts: Anterior thalamic radiation: ATR-L and ATR-R, Corticospinal tract: CST-L and CST-R, Cingulum (cingulate gyrus): CBC-L and CBC-R, Cingulum (hippocampus): CBH-L and CBH-R, Forceps major: FM-L and FM-R, Inferior fronto-occipital fasciculus: IFO-L and IFO-R, Inferior longitudinal fasciculus: ILF-L and ILF-R, Superior longitudinal fasciculus: SLF-L and SLF-R, Uncinate fasciculus: UF-L and UF-R, Superior longitudinal fasciculus (temporal part): SLFT-L and SLFT-R.

We observed a statistically significant correlation between mean FA and memory capacity ($r = 0.32$, $p < 0.001$, Figure R3.1A). To investigate whether memory capacity is a significant predictor of age in addition to the mean FA alone, we constructed two nested linear models with age as a dependent variable. In the first model, we fitted age as a function of mean FA, and then added the memory capacity as an additional predictor in the second model. We found that while mean FA was a good predictor of age ($R^2 = 0.39$, $p < 0.001$), adding the memory capacity resulted in a better model ($R^2 = 0.42$, $p < 0.001$). Using the likelihood-ratio test to

compare the two models, we found that adding the memory capacity to the model that only included mean FA resulted in a significantly more accurate model ($p < 10^{-6}$).

To understand further the relationship between mean FA and memory capacity in explaining age, we conducted a mediation analysis¹ (Figure R3.1B). This mediation analysis investigates whether the relationship between an independent variable (memory capacity) and a dependent variable (individual age) is mediated via a third variable (mean FA). Our results show that mean FA is a significant partial mediator of the relationship between memory capacity and age, where 53.2% of the total effect of memory capacity on age is mediated by mean FA.

Figure R3.1: Relationship between memory capacity and mean FA values. **A)** Associations between global memory capacity and mean FA values. Each circle represents an individual subject of the Cam-CAN cohort; the correlation was evaluated across the entire cohort. The orange line represents the best line of fit between the two variables. **B)** Mean FA values as a mediator of the relationship between memory capacity and age. The plot is showing effect estimates for the indirect (mediation), direct and total effect, where percentage of mediation was calculated as Indirect effect / Total effect * 100%.

Finally, we conducted a Partial Least Squares (PLS) regression analysis, where age was a dependent variable and memory capacity, mean FA, and individual FA values of the 20 tracts described above were included as predictors. PLS performs a linear decomposition of the predictor and predicted variable matrix into latent variables (LVs), which were optimized so that the covariance between the resulting predictor and predicted matrix components (called factors and loadings, respectively) is maximal². The contribution of each variable to the prediction was quantified via variable importance in the projection (VIP) scores, calculated as the sum of PLS weights over LVs, weighted by the variance explained by each LV. Variables were defined as significant contributors to the prediction based on a VIP score greater than 1³. Fig R3.2A shows that age was best explained by global memory capacity (VIP score = 2.02), followed by the FA values of the following tracts: Forceps major left (VIP score = 1.90) Inferior longitudinal fasciculus left (VIP score = 1.53), Cingulum (cingulate gyrus) right (VIP score = 1.36), Uncinate fasciculus right (VIP score = 1.30) and left (VIP score = 1.27), Inferior fronto-

occipital fasciculus right (VIP score = 1.14), and Cingulum (cingulate gyrus) left (VIP score = 1.04). Altogether, they were able to explain 63.84% of the variance in age (Fig R3.2B).

Figure R3.2: Relationship between age, memory capacity, and FA values of different tracts. **A)** The VIP scores for all measures used as predictors, where the significant contributors are shown in red (VIP > 1) and non-significant contributors are shown in gray. **B)** Amount of variance (for predicted and predictor variables) explained by the corresponding latent variables. Abbreviations: ATR-L/ATR-R: Anterior thalamic radiation left/right, CST-L/CST-R: Corticospinal tract left/right, CBC-L/CBC-R: Cingulum (cingulate gyrus) left/right, CBH-L/CBH-R: Cingulum (hippocampus) left/right, FM-L/FM-R: Forceps major left/right, IFO-L/IFO-R :Inferior fronto-occipital fasciculus left/right, ILF-L/ILF-R: Inferior longitudinal fasciculus left/right, SLF-L/SLF-R: Superior longitudinal fasciculus left/right, UF-L/UF-R: Uncinate fasciculus left/right, SLFT-L/SLFT-R: Superior longitudinal fasciculus (temporal part) left/right.

[1] Tingley D., *et al.* mediation: R Package for Causal Mediation Analysis. *J. Stat. Softw.* **59**(5), 1–38 (2014).

[2] Abdi H., *et al.* Partial least squares methods: partial least squares correlation and partial least square regression. *Comput. Toxicol. Volume II*, 549-579 (2013).

[3] Chong I. G., *et al.* Performance of some variable selection methods when multicollinearity is present. *Chemometr. Intell. Lab. Syst.* **78**(1-2), 103-112 (2005).

In summary, our additional analyses demonstrate that the measure of memory capacity is important for understanding connectome changes throughout aging and plays a significant role in age prediction. Specifically, it provides insights that are independent from, and complement those provided by the core measures of mean global FA and individual tract FA values.

We added these analyses in the revised version of our manuscript. The changes are as follows:

- In section **Results: The connectome of older individuals shows lower computational memory capacity** (pages 5-6), we added the following text:

“Moreover, we conducted additional analyses to assess whether topological measures or FA values explained the relationship between memory capacity and age. Our findings showed that, while these measures partially mediated the relationship, memory capacity provided independent information about connectome changes throughout aging (Supplementary Appendix: Section V, Section VI, and Section VII).”

- In the Supplementary Appendix, we added three new sections. Firstly, **Section V: Relationship between memory capacity and measures of network topology and brain structural integrity** (pages 5-7) details the relationship between memory capacity and global mean FA values. **Section VI: Mediation analysis of the relationship between memory capacity and age** (page 7) focuses on the mediation analysis in which mean FA values are investigated as mediator of the relationship between memory capacity and age. Finally, **Section VII: Relationship between age, memory capacity and FA values of individual tracts** (pages 7-8) summarizes the PLS analysis.
- Figure R3.1A and Figure R3.1B, together with their associated captions, were included as Figure S5D in section V and Figure S6B in section VI, respectively.
- Figure R3.2 and its caption was included as Figure S7 in section VII.

2. The authors use deterministic streamline tractography applied to multi-shell diffusion-weighted data and extracting the mean fractional anisotropy (FA). It's not clear (and I apologise if I missed it) whether the authors use FA simply to drive streamlines (as in a tensor framework) and use streamline count directly, or then go each connection and derive a measure based on the average FA along each tract? Ideally the authors would replicate their main finding by using a probabilistic approach as implemented in MrTrix 9or any similar toolbox) because deterministic approaches have a lot of false negative connections.

Response to comment: We apologize for the lack of clarity in our analysis. We used the diffusion tensor model to estimate the voxel-wise fiber orientation in the tractography analyses. The connectivity matrix was defined by the count of streamlines between each region.

Following the Reviewer's suggestion, we replicated our analysis by using a probabilistic approach to derive the anatomical connectomes. We used FSL bedpostX¹ for estimation of fiber orientation in the eddy-corrected and skull-stripped DWI images using a ball-and-stick model adjusted for multi-shell data. We also used the AAL2 atlas in order to define the seed and target masks for the tractography. Then, we built a whole-brain connectome for each individual by tracing a probabilistic distribution of streamlines from each region to all other regions using FSL probtrackx2². Finally, we defined the connectivity between two regions as the number of streamlines between them.

To evaluate whether the memory capacity of these probabilistic connectomes continued to be a reliable predictor of aging, we replicated the procedure to calculate memory capacity outlined in the main manuscript. Specifically, for each individual, we calculated the memory capacity at high-densities (21% to 30%) and calculated the area under the curve (AUC) for this range. We then used these AUC values as dependent variables in a linear model, with age as an independent variable and sex as a covariate. As shown in Figure R3.3A, memory capacity decreased with age, with age accounting for 18.8% of the variance in memory capacity ($p_{\text{age}} < 10^{-30}$). Our comparisons between groups of young and old (Figure R3.3B) further confirmed that older individuals (54-88 years old) exhibited significantly lower memory capacity

compared to younger individuals (18-53 years old). These findings demonstrate that memory capacity is a robust measure, with similar results obtained for the different methods used to calculate the individual anatomical connectomes.

Figure R3.3: Memory capacity for probabilistic tractography connectomes. **A)** Correlation between global memory capacity and age. The y-axis shows the AUC values of the memory capacity values calculated at the high-density range (21% to 30%). Orange area represents the 95% CI for predictions, the dashed lines indicate the best model fit, and the dots represent global memory capacity values for all individuals. **B)** Differences in global memory capacity between older (54-88 years old) and younger (18-53 years old) individuals. The boundaries of the boxes correspond to the 25th and 75th percentiles of the sample, with whiskers indicating non-outlier data.

[1] Jbabdi S., *et al.* Model-based analysis of multishell diffusion MR data for tractography: how to get over fitting problems. *Magn. Reson. Med.* 68, 1846–55 (2012).

2. Behrens T.E.J., *et al.* Probabilistic diffusion tractography with multiple fibre orientations: What can we gain? *Neuroimage* 34,144–55 (2007).

We added a summary of this additional analysis in our manuscript as follows:

- In section **Methods: Calculation of whole-brain anatomical connectivity networks** (page 18), we added the following text:

“To evaluate the robustness of our findings against different preprocessing pipelines, we replicated our results using probabilistic tractography to build the anatomical connectivity networks (Supplementary Appendix: Section XIV).”

- In the Supplementary Appendix, we added a new section, **Section XIV: Robustness against different preprocessing pipelines** (pages 26-27).
- Figure R3.3 and the associated caption was included as Figure S12 as part of this section.

3. Related to this point, are the connections in the reservoir algorithm (the matrix W on page 14) weighted or binary? If weighted, then I guess the number of edges will be the same per

each density threshold but the weights will generally become weaker with age (this speaks to my main concern, point 1, that FA might alone might be a simpler predictor).

Response to comment: We thank the Reviewer for this comment. Indeed, the binarization procedure was performed by normalizing the network by the number of edges at each density for all individuals. The weight of each connection was preserved in each network, and the memory capacity was calculated on such weighted networks. We would like to highlight that the weights were normalized prior to calculating the memory capacity, such that the largest singular value of each network was set to one. This normalization removed the effect of decreasing weights with age and ensured that this effect was also removed while evaluating the memory capacity.

To demonstrate this, we calculated a linear model to understand how much of the variance in age can be explained by the average network weight, calculated after the normalization of the network. Analogous to the process of evaluating high density memory capacity, we calculated average weights for each network in the density range 21% to 30% and evaluated the area under the curve across this range. Our findings showed that the average weight can explain only a modest amount of variance in age ($R^2 = 4.5\%$, $p < 0.001$, Figure R3.4A), which is lower than the one that can be explained by the memory capacity (10.9% explained variance, $p_{\text{age}} < 10^{-15}$). To further reinforce this point, we performed a mediation analysis to understand whether average weight mediates some of the effects of memory capacity on age. Our results, shown in Figure R3.4B, found that average normalized weight mediates only a marginal part of the relationship between memory capacity and age (8.8%, $p = 0.04$). Therefore, the memory capacity provides information about the network changes throughout aging which is complementary to the one provided by the average weight.

Figure R3.4: Relationship between memory capacity, average network weight and age. **A)** Correlation between average network weight and age. The y-axis shows the AUC values of the average weight values calculated at the high-density range (21% to 30%). The weight was calculated after normalizing the network so that its largest singular value is equal to 1. Orange area represents the 95% CI for predictions, the dashed lines indicate the best model fit, and the dots represent global memory capacity values for all individuals. **B)** Average weight as a

mediator of the relationship between memory capacity and age. The plot is showing effect estimates for the indirect (mediation), direct and total effect, where percentage of mediation was calculated as Indirect effect / Total effect * 100%.

We modified our manuscript to add these analysis. The changes are as follows:

- In section **Results: The connectome of older individuals shows lower computational memory capacity** (pages 5-6), we added the following text:

“Moreover, we conducted additional analyses to assess whether topological measures or FA values explained the relationship between memory capacity and age. Our findings showed that, while these measures partially mediated the relationship, memory capacity provided independent information about connectome changes throughout aging (Supplementary Appendix: Section V, Section VI, and Section VII).”

- In the Supplementary Appendix, we added two new sections. Firstly, **Section V: Relationship between memory capacity and measures of network topology and brain structural integrity** (pages 5-7) details the relationship between memory capacity and average network weight. **Section VI: Mediation analysis of the relationship between memory capacity and age** (page 7) focuses on the mediation analysis where we investigated whether the average network weight value is significant mediator relationship between memory capacity and age.
- Figure R3.4B, together with its associated caption, was included as Figure S6A in section VI, respectively.

Minor:

1. “Since the initial part of the curve ($\tau = 1$ to $\tau = 5$) did not change with aging, we measured \ the memory capacity between delays $\tau = 6$ to $\tau = 35$ (dark-gray shaded area).” – this sounds circular – i.e. the choice of time window was made after already looking at the predicted outcome.

Response to comment: The choice to evaluate memory capacity between the delays $\tau = 6$ to $\tau = 35$ was to emphasize that the ability of brain reservoirs to memorize temporal information across the lifespan depends on how much delayed the information is. Since our initial analyses pointed out that immediate signals (delays $\tau = 1$ to $\tau = 5$) were not affected by age-related changes, we focused on the memory capacity derived using delayed signals as a marker for aging.

However, we would like to point out that our results remained valid also when we considered the more traditional definition of memory capacity and used all delays between $\tau = 1$ to $\tau = 35$ to evaluate it. To show this, we repeated several of our analyses and showed that similar conclusions can be drawn even if we use this definition of memory capacity. Namely, Figure R3.5 shows that there was a significant decrease in the memory capacity of old individuals, where the differences increased as the density of the network increased. When evaluating the area under the curve for low (2%-10%), medium (11%-20%) and high (21%-30%) density ranges, we found that memory capacity could explain 4%, 6.9%, and 10.6% of the variance in age, respectively. These values were similar to the results presented in the original manuscript, which were 3.98%, 6.87%, and 10.9% for the three density ranges, respectively.

Figure R3.5: Memory capacity in aging when memory capacity is defined for delays $\tau = 1$ to $\tau = 35$. **A)** Differences in global memory capacity between older (54-88 years old) and younger (18-53 years old) individuals. The plot shows the 95% confidence intervals (CI) in orange, while the circles represent the differences in global memory capacity as a function of network density; the significant differences fall outside the CI. Correlation between global memory capacity and age at **B)** low, **C)** medium, and **D)** high densities. In all cases, the y-axis shows the AUC values of the memory capacity values calculated across all density values in each category. Orange areas represent the 95% CI for predictions, dashed lines indicate the best model fit, and the dots represent global memory capacity values for all individuals. Boxplots display global memory capacity values for young and old groups, where the boundaries of the box correspond to the 25th and 75th percentiles of the sample with whiskers indicating non-outlier data.

2. Why the same time series to all nodes? And why then randomly scaled by W (I assume this is what happens); why not just different random time series to each input node? Is this standard. I'm just lacking the background here.

Response to comment: We understand the Reviewer's comment. In the current study, we applied the same time series for all nodes because this is the standard approach for reservoir computing^{1,2}. However, it might be interesting to explore the possibility of applying different time series to different nodes in future works.

[1] Lukoševičius, M. & Jaeger, H. Reservoir computing approaches to recurrent neural network training. *Comput. Sci. Rev.* **3**, 127–149 (2009).

[2] Jaeger, H. Short term memory in echo state networks. (2001).

We revised the manuscript in order to make this point clearer:

- In section **Methods: Reservoir computing** (page 14), we added the following text:

“Following the standard approach for reservoir computing, we fed the same time series to all reservoir nodes ...”

3. “where tanh units” -> “were”

Response to comment: We apologize for the mistake, we corrected it in the revised version of the manuscript.

1. Rubinov, M. (2023). Circular and unified analysis in network neuroscience. *Elife*, 12, e79559.

Reviewer #3 (Remarks on code availability):

The code and data availability statements are appropriate. I checked the links to the code including the github repository which appear sound

Response to comment: We thank the reviewer. In order to facilitate even easier access to the codes, we are now hosting the code at: <https://github.com/braph-software/MemoryCapacity>.

Reviewer #1 (Remarks to the Author):

Dear Editor,

I would like to thank the authors for conducting the complementary experiments and addressing the questions and points I raised earlier. The authors' detailed and insightful responses and adjustments to the structure and narrative of the manuscript have clarified most of my concerns.

Response to comment: We thank the Reviewer. We are pleased that our additional analyses clarified most of the Reviewer's concerns regarding the manuscript.

However, there is still one major issue that needs to be addressed before considering this work for publication. Specifically, the method for linking genetic data and the observed age-related differences in memory capacity requires further clarification and justification.

1. Motivation:

- While the motivation has been partially explained in the response to my previous comments, it would be beneficial if the authors could elaborate on the relevance of the method in the context of their work as well as in the broader context of memory decline during the aging process. This would provide a clearer understanding of the importance and potential impact of this study.

2. Method Justification and Explanation:

- I acknowledge that regional memory capacities (MCs) reflect connectomic features and their variation during aging. However, linking genetic data with these “derived” features (i.e., differences in regional MCs that are products of another machine learning model) seems overly complex and potentially unreliable without further independent assessment and justification. Linking genetic data with “direct” features extracted from connectomic and functional data (as in references listed by authors in their response to reviewers) is typically easier to justify and interpret.

- The authors cite several studies that use methods to characterize associations between gene transcription profiles and direct features extracted from connectomes and functional networks. These studies typically use multivariate methods, such as partial least squares (PLS), to identify associations between sets of response variables and predictor variables. The method in the current manuscript differs from these established methods. The manuscript uses a different approach introduced in [1], where XGboost is employed to classify hubs. Here, the XGBoost model is employed as a regressor, which is not sufficiently justified. I understand that the size of the predictor matrix (i.e., 1158 cortical samples x 10027 genes) might have motivated using advanced machine learning models, however, the spatial resolution of the response set is much lower (i.e., only 94 nodes compared to 1158 cortical samples) and the applied up-scaling/super-resolution technique would play a critical role, specifically for the regression (compared to a classification problem in [1]).

- In the manuscript, the dimension of the predictor matrix (1158 cortical samples × 10027 genes) is only mentioned in Figure 2, while the response vector's dimension remains ambiguous. Based on Figure 2, it seems that the response vector contains differences in memory capacity, suggesting significant upscaling (~12 times larger). The explanation in Supplementary Information IX indicates that memory capacity (MC) values are assigned to

cortical samples based on spatial coordinates using the AAL atlas. This could be done through value copying, linear/nonlinear extrapolations, or advanced super-resolution methods. It should be clarified.

- The Allen Human Brain Atlas (AHBA) uses a different anatomical reference, not the AAL atlas. This discrepancy introduces uncontrolled parameters to the model and interpretations.

- Despite using a sophisticated model like the XGBoost regressor, the prediction accuracy remains modest. Given the current parcellation, the XGBoost regressor, the modest prediction, and the use of two different anatomical atlases, the reliability of the method and these findings remain questionable.

Conclusion and Recommendations

While I remain critical of the method, its findings, and interpretations, if the authors wish to retain this part of the manuscript, I suggest the following:

- Summarize the method in a simpler and clearer manner to enhance reader comprehension without needing to consult additional figures and references.
- Provide a stronger justification for the use of “derived” features and the chosen machine learning approach in the context of this study.

Thank you again for your efforts. I look forward to the final version of your paper.

Ref:

[1] Xu, Zhilei, et al. "Meta-connectomic analysis maps consistent, reproducible, and transcriptionally relevant functional connectome hubs in the human brain." *Communications Biology* 5.1 (2022): 1056.

Response to comment: We agree with the Reviewer and acknowledge that the Reviewer has raised valid points regarding the limitations of the method we used to link memory capacity to genetic data. Since this was a supplementary analysis that does not directly affect the conclusions of our manuscript, and given the Reviewer’s previous suggestion that its removal would improve the flow and clarity of the manuscript, following the original Reviewer’s suggestion, we have now decided to remove this analysis from the manuscript. We believe that this adjustment will improve the manuscript by maintaining a more focused message and reducing complexity.

Specifically, we made the following changes:

- In the **Abstract**, we removed the sentence “*being affected by genes with a role in inter-neuronal inhibitory communication*”. The modified abstract is reproduced below:

“Memory is a crucial cognitive function that deteriorates with age. However, this ability is normally assessed using cognitive tests instead of the architecture of brain networks. Here, we use reservoir computing, a recurrent neural network computing paradigm, to assess the linear memory capacities of neural-network reservoirs extracted from brain anatomical connectivity data in a lifespan cohort of 636 individuals. The computational memory capacity emerges as a robust marker of aging, being associated with resting-state functional activity, white matter integrity, locus coeruleus signal intensity, and cognitive performance. We replicate our

findings in an independent cohort of 154 young and 72 old individuals. By linking the computational memory capacity of the brain network with cognition, brain function and integrity, our findings open new pathways to employ reservoir computing to investigate aging and age-related disorders.”

- We removed the following sentence from the **Introduction** (page 3):

“Furthermore, we use transcriptomic analysis to identify the genes affecting the age-related differences in computational memory capacity, and demonstrate a link between the computational memory capacity and synaptic communication between inhibitory neurons.”

- We removed the section “**Regions showing age-related changes in memory capacity overlap with regions showing hyper-excitability in aging**” (pages 6-7).
- We removed the following sentence from the **Discussion** (page 8):

“We provide a biological interpretation of memory capacity by linking it to genes involved in synaptic communication processes in inhibitory neurons.”

- We removed the paragraph discussing the results obtained using the transcriptomics analysis in the **Discussion** (pages 9-10).
- We removed the limitations part of the Manuscript that is related to the transcriptomics analysis (pages 11-12).
- We removed the following sentence from the **Discussion** (page 11):

“Furthermore, our findings suggest a genetic basis for memory capacity, showing that it is associated with the levels of inhibition in the brain.”

- In the **Methods**, we removed the section “**Characterizing the transcriptomic basis of age-related regional memory capacity differences**”.
- We removed the relevant sentences from the **Data availability** and **Code availability** sections.
- We removed Figure 2 from the Manuscript and adjusted the references to other figures accordingly.
- We removed “**Section IX: Methodological considerations of transcriptomics analysis**” from the Supplementary Appendix.

Reviewer #1 (Remarks on code availability):

The code only contains a pipeline to compute memory capacity in the reservoir computing framework.

The source codes for linking genetic data and the observed age-related differences in memory capacity are missing.

Response to comment: Following the removal of the analyses that linked age-related differences in memory capacity to the genetic data, we do not need to add the code for this analysis to the GitHub repository.

Reviewer #3 (Remarks to the Author):

The authors have responded very thoroughly and with impressive additional analyses:

Studying the partial influence of age/FA as a mediator

Replicating with probabilistic tractography

Clarifying methodological steps and parameter choices

This is an interesting and important contribution to the field.

Michael Breakspear

Response to comment: We would like to thank the Reviewer for their time, effort, and valuable comments throughout the peer-review process.

Cross-revision on Reviewer #2's behalf:

R2.1. Normalization is commonly applied to adjust the largest eigenvalue ($\rho = 1.0$) in order to satisfy the echo state property in reservoir computing. However, this choice directly affects the memory capacity of the system. The authors conducted supplementary experiments and reported age-prediction accuracy in relation to the memory capacity (MC) vector for both $\rho = 0.96$ and $\rho = 1.0$, showing comparable results. In their table, they also performed a grid search, reporting the minimum and maximum global MC values for various ρ settings. However, they did not address the possibility that each individual's structural network may have an optimal ρ value, specifically tailored to maximize individual MC. It would be beneficial for the authors to visualize the average MC per group, as well as between-group differences, by adjusting ρ to its optimized value for each individual, and to illustrate trends similar to those depicted in Figure R1.3 (in response to Reviewer #1). Additionally, reporting the vector of ρ_{opt} would provide insights into whether different age groups exhibit optimal ρ values within distinct intervals.

Response to comment: We thank the Reviewer for this comment. Following the Reviewer's suggestion, we recalculated the memory capacity by adjusting the ρ parameter for each individual to its optimal value (ρ_{opt}) that maximizes the memory capacity. Our findings indicate that most participants ($N = 443$, 69.7%) achieved maximal memory capacity at $\rho_{\text{opt}} = 0.96$, consistent with our earlier results. Additionally, 2 participants (0.3%) showed optimal performance at $\rho_{\text{opt}} = 0.94$, 182 participants (28.6%) at $\rho_{\text{opt}} = 0.95$, and 9 participants (1.4%) at $\rho_{\text{opt}} = 0.97$.

To assess the age-related differences in ρ_{opt} , we divided the cohort into young and old individuals using age thresholds ranging from 39 to 70 years. We found no significant differences between the age groups (p values ranging from 0.29 to 0.96), suggesting that there is no relationship between age and the optimal ρ value. Additionally, we also compared the optimal memory capacity (calculated using the ρ_{opt} values for each individual) between young and old individuals using the same range for age thresholds (Figure R2.1). Our results showed that the older individuals had lower optimal memory capacity when compared to young ones across the complete range of age thresholds (Figure R2.1A and Figure R2.1B), which was significant for all thresholds ($p_{\text{max}} < 10^{-4}$). These analyses show that memory capacity exhibits consistent trends across different normalization procedures for the connectivity networks.

Figure R2.1: Optimal memory capacity for each individual as a function of the age threshold used to define young and old individuals. A) Average optimal memory capacity for young (orange circles) and old (red circles) individuals as a function of the age-threshold used to categorize subjects into these groups. The error bars represent one standard error of the mean for each group. **B)** Between-group differences (calculated as old minus young) as a function of the age-threshold.

We revised the Supplementary Appendix to add these additional analyses:

- In **Section XII: Normalization procedure for the whole-brain anatomical networks** (pages 12-13), we added the following text (please note that we use the notation s_{SVD} to indicate the maximal singular value in order to achieve consistency with the rest of Section XII):

“To account for the possibility that each individual connectome can reach optimal memory performance at different s_{SVD} values, we recalculated the memory capacity by normalizing each connectome by its optimal s_{SVD} value (denoted by s_{SVD}^{opt}). Our results indicate that the majority of the participants ($N = 443$, 69.7%) achieved optimal memory capacity at a value of 0.96, consistent with Table S2. Furthermore, 2 participants (0.3%) showed optimal performance at $s_{SVD}^{opt} = 0.94$, 182 participants (28.6%) at $s_{SVD}^{opt} = 0.95$, and 9 participants (1.4%) at $s_{SVD}^{opt} = 0.97$.

We investigated the age-related differences in the optimal memory capacity and s_{SVD}^{opt} by dividing the cohort into young and old individuals using age thresholds from 39 to 70 years (Section IV). We observed no significant differences in s_{SVD}^{opt} between the age groups (p values ranging from 0.29 to 0.96), suggesting no relationship between age and the singular value that leads to optimal performance. However, when comparing the optimal memory capacity across these age thresholds, we found that old individuals consistently had lower memory capacity than young individuals (Fig. S11). These results align with those shown in Fig. S4, indicating that memory capacity exhibits similar trends across different normalization procedures.”

- Figure R2.1 was added as Figure S11 in Supplementary Section XII.

R2.3 The authors provide a clear explanation of the procedure and acknowledge the reviewer's point regarding the potential issue of spatial autocorrelation in this analysis. However, their method justification—stating that the method “has received widespread validation in many studies that associate previously known genes with spatial distributions of several measures of brain structure and function across various disorders”—is not entirely convincing. Without further independent validation and stronger justification, linking genetic data to the RC memory performance which is, in turn, affected by RC implementation, low-resolution parcellation, and RC parameter set (e.g., ρ) may introduce unnecessary complexity and reduce reliability.

Response to comment: We thank the Reviewer for this comment and for recognizing that the procedure for linking genetic data to RC memory performance is more clearly explained in our revised manuscript. However, considering the limitations of this method, the fact that it was a secondary analysis without an impact on the main conclusions, and the earlier suggestion of Reviewer 1 that its removal could improve the manuscript by streamlining its message and removing complexity, we have decided to exclude this analysis from the revised manuscript. This analysis has been removed from both the main manuscript and the supplementary appendix as follows:

- We removed the sentence “*being affected by genes with a role in inter-neuronal inhibitory communication*” from the **Abstract**. The modified abstract is shown below:

“Memory is a crucial cognitive function that deteriorates with age. However, this ability is normally assessed using cognitive tests instead of the architecture of brain networks. Here, we use reservoir computing, a recurrent neural network computing paradigm, to assess the linear memory capacities of neural-network reservoirs extracted from brain anatomical connectivity data in a lifespan cohort of 636 individuals. The computational memory capacity emerges as a robust marker of aging, being associated with resting-state functional activity, white matter integrity, locus coeruleus signal intensity, and cognitive performance. We replicate our findings in an independent cohort of 154 young and 72 old individuals. By linking the computational memory capacity of the brain network with cognition, brain function and integrity, our findings open new pathways to employ reservoir computing to investigate aging and age-related disorders.”

- We removed the following sentence from the **Introduction** (page 3):

“Furthermore, we use transcriptomic analysis to identify the genes affecting the age-related differences in computational memory capacity, and demonstrate a link between the computational memory capacity and synaptic communication between inhibitory neurons.”

- We removed the section “**Regions showing age-related changes in memory capacity overlap with regions showing hyper-excitability in aging**” (pages 6-7).
- We removed the following sentence from the **Discussion** (page 8):

“We provide a biological interpretation of memory capacity by linking it to genes involved in synaptic communication processes in inhibitory neurons.”

- We removed the paragraph discussing the results obtained using the transcriptomics analysis in the **Discussion** (pages 9-10).

- We removed the limitations part of the Manuscript that is related to the transcriptomics analysis (pages 11-12).
- We removed the following sentence from the **Discussion** (page 11):

“Furthermore, our findings suggest a genetic basis for memory capacity, showing that it is associated with the levels of inhibition in the brain.”

- In the **Methods**, we removed the section “**Characterizing the transcriptomic basis of age-related regional memory capacity differences**”.
- We removed the relevant sentences from the **Data availability** and **Code availability** sections.
- We removed Figure 2 from the Manuscript and adjusted the references to other figures accordingly.
- We removed “**Section IX: Methodological considerations of transcriptomics analysis**” from the Supplementary Appendix.

Reviewer #1 (Remarks to the Author):

Dear Editor,

I appreciate the authors' efforts in making the necessary adjustments to ensure the manuscript maintains its focus on the central idea, as well as the primary analyses and findings. The revisions have greatly enhanced the clarity and impact of the work.

I have no further comments at this stage. The manuscript is well-structured and effectively presents the authors' excellent research.

Best regards,

Response to comment: We would like to thank the Reviewer for their time, effort and constructive comments. We are pleased that our efforts were able to clarify the Reviewer's concerns regarding the manuscript.